# Context dependent effects of ascorbic acid treatment in TET2 mutant myeloid neoplasia

Yihong Guan [1], Edward F. Greenberg[1,2], Metis Hasipek[1], Shi Chen[3,4], Xiaochen Liu[3], Cassandra M. Kerr[1], Daniel Gackowski [5], Ewelina Zarakowska[5], Tomas Radivoyevitch [1], Xiaorong Gu[1], Belinda Willard[6], Valeria Visconte[1], Hideki Makishima [1,7], Aziz Nazha[1,2], Mridul Mukherji[8], Mikkael A. Sekeres[2], Yogen Saunthararajah [1], Ryszard Oliński[5], Mingjiang Xu[3,4], Jaroslaw P. Maciejewski[1,2✉] & Babal K. Jha [1✉]

Loss-of-function TET2 mutations ($TET2^{MT}$) are common in myeloid neoplasia. TET2, a DNA dioxygenase, requires 2-oxoglutarate and Fe(II) to oxidize 5-methylcytosine. $TET2^{MT}$ thus result in hypermethylation and transcriptional repression. Ascorbic acid (AA) increases dioxygenase activity by facilitating Fe(III)/Fe(II) redox reaction and may alleviate some biological consequences of $TET2^{MT}$ by restoring dioxygenase activity. Here, we report the utility of AA in the prevention of $TET2^{MT}$ myeloid neoplasia (MN), clarify the mechanistic underpinning of the TET2-AA interactions, and demonstrate that the ability of AA to restore TET2 activity in cells depends on N- and C-terminal lysine acetylation and nature of $TET2^{MT}$. Consequently, pharmacologic modulation of acetyltransferases and histone deacetylases may regulate TET dioxygenase-dependent AA effects. Thus, our study highlights the contribution of factors that may enhance or attenuate AA effects on TET2 and provides a rationale for novel therapeutic approaches including combinations of AA with class I/II HDAC inhibitor or sirtuin activators in $TET2^{MT}$ leukemia.

[1] Department of Translational Hematology and Oncology Research, Taussig Cancer Institute, Cleveland Clinic, Cleveland, OH, USA. [2] Leukemia Program, Department of Hematologic Oncology and Blood Disorders, Taussig Cancer Institute, Cleveland Clinic, Cleveland, OH, USA. [3] Department of Cell System & Anatomy, University of Texas Health at San Antonio, San Antonio, TX 78229, USA. [4] Sylvester Comprehensive Cancer Center, Department of Biochemistry and Molecular Biology, University of Miami Miller School of Medicine, Miami, FL, USA. [5] Department of Clinical Biochemistry, Faculty of Pharmacy, Collegium Medicum in Bydgoszcz, Nicolaus Copernicus University in Toruń, 85-095 Bydgoszcz, Poland. [6] Proteomics and Metabolomics Core, Lerner Research Institute, Cleveland Clinic, Cleveland, OH, USA. [7] Department of Pathology and Tumor Biology, Graduate School of Medicine, Kyoto University, Kyoto, Japan. [8] UMKC School of Pharmacy, Kansas City, MO, USA. ✉email: maciejj@ccf.org; jhab@ccf.org

TET2 is frequently affected by hypomorphic missense mutations and loss-of-function (LOF) nonsense/frameshift-truncating mutations[1,2]. TET2 is a $Fe^{2+}$ catalyzed 2-oxoglutarate (αKG) dependent DNA-dioxygenase that progressively oxidizes 5-methylcytosine (5mC) in DNA to 5-hydroxymethylcytosine (5hmC), 5-formylcytosine (5fC), and 5-carboxylcytosine (5caC)[3]. These TET-dependent 5mC-DNA oxidation products (TDOP) are replaced by C via base excision repair that regulate transcription profile determining cell lineage fate, proliferation and survival[3–7]. In addition, DNA demethylation are also achieved due to the inability of methyltransferase to copy hydroxylated methylation (5hmC) marks during replication.

TET2 mutations ($TET2^{MT}$) are common in myeloid neoplasia (MN), particularly chronic myelomonocytic leukemia and myelodysplastic syndrome (MDS)[1,8,9], and in T/B -cell lymphomas[10]. $TET2^{MT}$ are also found in the healthy elderly individuals with clonal hematopoiesis of indeterminate potential (CHIP)[11–16]. TET2 deficiency alone produces an initially benign phenotype characterized by a long disease latency and incomplete penetrance[1,9,15]. However, $TET2^{MT}$ in CHIP suggest that they are often a first step toward progression to MN[17]. The incidence of both CHIP and $TET2^{MT}$ neoplasia increases with age: up to 60% of MDS in octogenarian harbor $TET2^{MT}$[1,9,18]. $Tet2^{-/-}$ and $Tet2^{+/-}$ mice develop an initially mild myeloproliferative syndrome[19]. Acceleration of this process by radiation suggests that additional genetic hits may lead to a higher mutational burden enabled by a primary TET2 lesion[20].

Ascorbic acid (AA) enhances the activity of TET2 likely by reducing catalytic site Fe(III) to Fe(II)[21,22]. Simultaneous 2-oxoglutarate decarboxylation provides two additional electrons needed for dioxygenase activity. AA may thus improve the function of intact TET2 in cases with heterozygous $TET2^{MT}$, or increase compensatory enzymatic activity of TET1/3 in cases with biallelic TET2 inactivation. Such effects may restore/improve hydroxymethylation, and potentially reverse the epigenetic consequences caused by TET2 deficiency. In agreement with these theoretical predictions, recent reports have demonstrated the biologic activity of AA on $TET2^{MT}$ hematopoietic cells in vitro, in $Tet2^{MT}$-mediated leukemia models[23–26], and in melanoma[27]. Interestingly, patients with leukemia have lower serum AA concentrations compared to age-matched controls[28], and their AA levels continue to decline with age[29,30]. However, this age-dependence of serum AA levels have not been observed in healthy individuals[31]. The effect of AA in preventing leukemogenesis was recently reported[23,26]; these studies relied heavily on bone marrow (BM) transplantation models, and not enough is known about the effect of AA on genetic models of the evolution of MN. A recent case report[32] did suggest that an acute supraphysiological dose of AA may benefit $TET2^{MT}$ AML as a single agent. However, it remained unclear what is the contribution of TET dioxygenase activation and what are the TET2-dioxygenase independent effects of AA. In addition, other factors may affect TET2 activity and therefore either enhance or attenuate AA effects. For instance, post-translational context-dependent acetylation and deacetylation of TET2 lysine residues may enhance or reduce TET2 stability and activity[33,34]. Acetylation of TET2 N-terminal lysine residues prevents proteasomal degradation[34]. In contrast, TET2 catalytic domain lysine residues deacetylation by sirtuins may increase its activity.

In this study, we investigated the long term impact of AA treatment in the prevention of MN evolution in murine model and the mechanistic effects of AA on TET2 mutant human myeloid cells derived from the MN patients. In addition, we demonstrate that the catalytic domain lysine acetylation and missense mutations that closely mimics lysine acetylation have significant effect on TET dioxygenase activity. The loss of function in TET2 caused by catalytic domain lysine acetylation or missense mutation cannot be restored by AA treatment. However, addition of chemical probes regulating lysine acetylation amplifies AA mediated TET-dioxygenase activity in leukemia cells. Our study further rationalizes the targeted therapeutic application of AA alone or in combination with other agents in $TET2^{MT}$ myeloid leukemia.

## Results

**Long-term oral AA treatment slows myeloproliferation in Tet2$^{+/-}$ mice.** In vivo effects of AA in preventing leukemogenesis were established in transplant models of Tet2 knockout leukemia[23,26], but remained unexplored in a direct genetic model of Tet2-deficiency. We therefore, performed several key experiments and probed the effects of AA treatment in $Tet2^{mt}$ murine model. We cultured mononuclear cells purified from BMs and spleens of $Tet2^{+/+}$, $Tet2^{+/-}$ and $Tet2^{-/-}$ mice in the presence or absence of AA and measured TDOP levels. AA increased TET-activity in mononuclear cells derived from mice with the 3 genotypic configurations as reflected in increased TDOP levels (Supplementary Figs. 1a–e and Supplementary Data 1) and decreased colony numbers (Supplementary Fig. 1f). To further explore the in vivo therapeutic efficacy of long-term AA supplementation in preventing disease evolution we treated $Tet2^{+/-}$ mice (3.3 g/L, AA drinking water) for one year and analyzed the disease progression. Control $Tet2^{+/-}$ mice receiving water, spontaneously developed splenomegaly and increased proportion of Gr-1$^+$/Mac-1$^+$ myeloid lineage cells in the peripheral blood (PB), spleen and BM. This development was partially reversed with AA supplement (Fig. 1a–e and Supplementary Fig. 1g) as observed in lower spleen weight (Fig. 1c), monocyte counts (Fig. 1b) and myeloid lineage proportions (Fig. 1d–e). There were no significant effects of AA treatment on total counts of WBC, RBC, and neutrophils in PB or LSK/LK and Gr-1+/Mac-1+ myeloid lineage cell proportions in BM (Supplementary Fig. 1g).

We also exposed both cohorts of WT and $Tet2^{+/-}$ mice to sublethal ionizing radiation (IR), and maintained them with or without 3.3 g/L oral AA supplementation in drinking water. IR exposure had no effect on 1-year overall survival (OS) to WT mice on the other hand $Tet2^{+/-}$ mice had only a 40% 1-year OS (Fig. 2a, b). Oral AA treatment improved the 1-year OS of IR exposed $Tet2^{+/-}$ mice to 70%, and reduced BM myeloid-lineage cells fraction by ~20% in average, compared to the untreated IR-$Tet2^{+/-}$ mice (Fig. 2c).

To mitigate the impact of mice producing endogenous AA via L-gulonolactone oxidase (Gulo), we crossed homozygous $Gulo^{-/-}$ mice with Tet2 knockout mice to generate hybrid $Tet2^{+/-}$;$Gulo^{-/-}$ mice (Fig. 2d). $Gulo^{-/-}$ mice are unable to synthesize AA, which is essential for the survival of mice[35]. Therefore, in our experiment, we supplied the mice with minimal sustenance dose of AA (0.033 g/L) in drinking water. AA treatment significantly increased 5hmC/5mC, 5fC/5mC, and 5caC/5mC (Fig. 2e–g). We observed a dose dependent increase in the survival of $Tet2^{+/-}$;$Gulo^{-/-}$ mice upon AA treatment. The median survival increased 36 days for 0.33 g/L and 116 days for 3.33 g/L AA supplementation compared to the minimal dose of AA supplementation (0.033 g/L) required for the sustenance of these mice, however, this effect does not achieve statistical significance in our experimental settings (Fig. 2h, i).

**Impact of AA treatment on TET2 activation and proliferation.** Partial loss of TET activity due to hypomorphic $TET2^{MT}$ drives myeloproliferation in patients, and restoring TET2 function by overexpression of TET2 or by AA treatment should inhibit this proliferation. We next studied the ability of AA to increase TET2 activity in experimentally-generated TET2-overexpressing

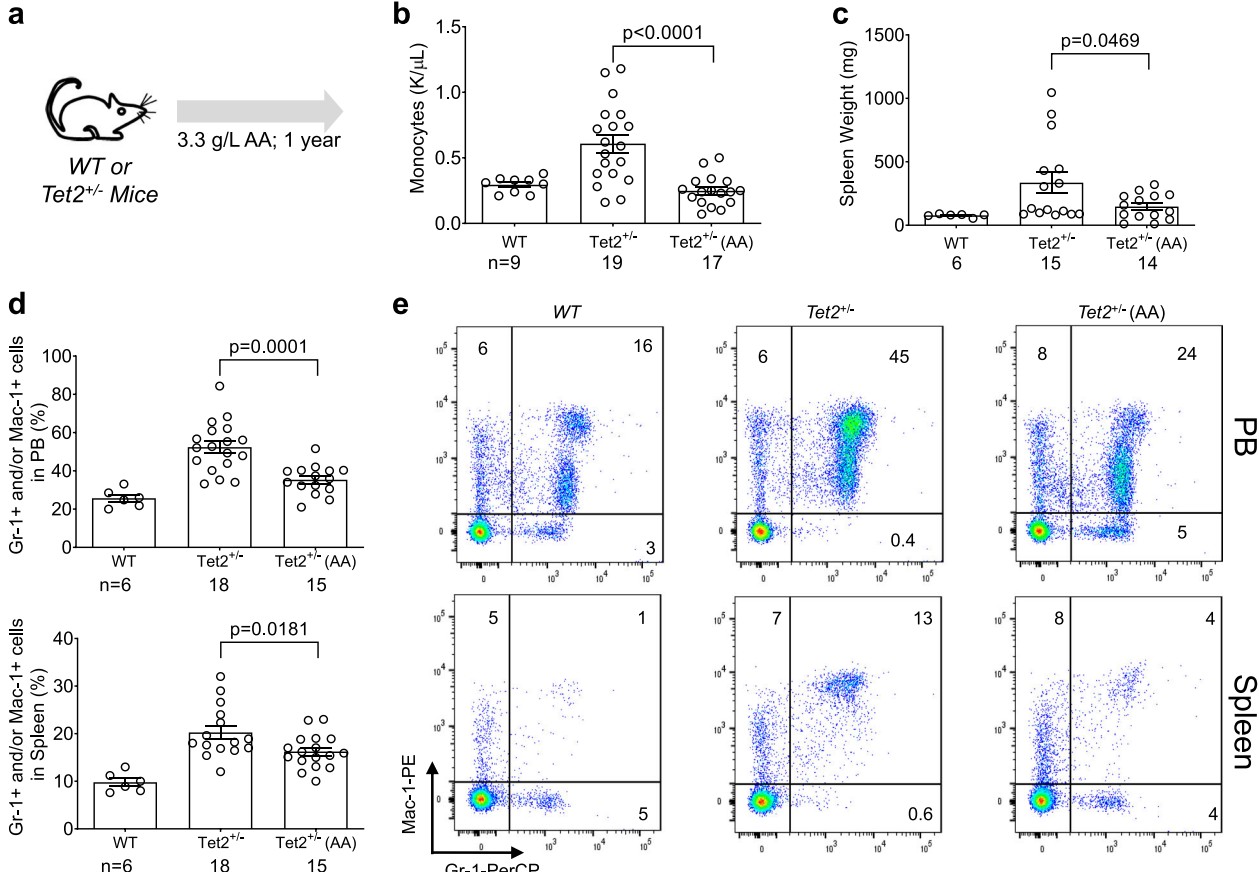

**Fig. 1 Long-term Oral AA Treatment Slows Myeloproliferation in Tet2$^{+/-}$ Mice. a** A 6–8 week-old *Tet2$^{+/-}$* mice were treated for 1 year with 3.3 g/L AA or water, animals were then sacrificed and analyzed for Monocyte counts (**b**), Spleen weight (**c**), and Gr1$^+$/Mac1$^+$ proportions (**d**) in peripheral blood (PB, top panel) and spleen (bottom panel). **e** Representative results of flow cytometric analyses of monocytic/granulocytic cells (Gr1$^+$/Mac1$^+$) in PB and spleen of representative WT and *TET2$^{+/-}$* mice. The number in each quadrant is the percentage of cells. **b**, **d**. Data are shown with group mean with SEM; statistical significance (*p* values) from two tailed t-test are indicated.

(TET2$^{OE}$) and knockdown (TET2$^{KD}$) cell lines. For this purpose, we ectopically expressed TET2 into both *TET2$^{WT}$* (MEG-01 and CMK) and *TET2$^{MT}$* (HEL) leukemia cell lines (Supplementary Data 2). *TET2$^{OE}$* cells have significantly higher TET-activity (up to eightfold) compared to the vector control cells (Fig. 3a and Supplementary Fig. 2a). *TET2$^{OE}$* cells grew 2-4-fold slower than control cells correlating with increased TDOPs (Fig. 3a–d and Supplementary Fig. 2a). Similar growth perturbations were observed with AA treatment for control cells (Fig. 3b–d). In contrast, transduction of TET2 shRNA consistently reduced *TET2* mRNA levels by 80–90% in K562 and MOLM13 cells and were associated with an approximately twofold decrease in 5hmC/5mC and 5fC/5mC (Fig. 3e, f and Supplementary Fig. 2b-c). Treatment of these cells with pharmacologic doses of AA increased 5hmC/5mC and 5fC/5mC ratios in both *TET2$^{WT}$* (3-5-fold) and *TET2$^{KD}$* (3-4-fold) (Fig. 3e, f and Supplementary Figs. 2b-c). AA treatment significantly reduced the growth of *TET2$^{KD}$* cells as well as the parental cells. However, the AA impact on the growth of *TET2$^{KD}$* cells were more pronounced (Fig. 3e–h and Supplementary Figs. 2d-e). Colony forming cultures were performed with purified human CD34$^+$ cells transduced with *TET2* shRNA or scrambled control. Following transduction, CD34$^+$ cells were grown in methylcellulose media with or without AA supplementation. While *TET2* shRNA increased colony formation, AA treatment reduced the growth of both control and *TET2$^{KD}$* CD34$^+$ cells (Fig. 3i, j). We also studied AA-mediated up modulation of TET2 activity in primary BM derived mononuclear

cells isolated from MN patients with *TET2$^{WT}$* or *TET2$^{MT}$* (Supplementary Data 3). In contrast to the genetically engineered mouse cells and human leukemic cell lines, AA only modestly increased TDOP levels in *TET2$^{WT}$* while *TET2$^{MT}$* showed mixed results. Even 250 μM AA treatment did not affect TDOP levels in a significant manner. Overall TET activity as reflected in 5hmC/5mC among *TET2$^{MT}$* cells were lower than in *TET2$^{WT}$* control (Fig. 3k). AA treatment had statistically insignificant effect on 5hmC/5mC on *TET2$^{MT}$* patients but the modest effect observed in *TET2$^{WT}$* patients were statistically significant as assessed using quantitative mass spectrometry (Fig. 3k, l). To better understand the mechanism of AA mediated TET2 activation and its utility in *TET2$^{MT}$* MN patients we further analyzed AA's effect on the dioxygenase activity of TET2.

**AA binds to TET2$^{CD}$ and increases its activity**. To better understand the mechanism of AA-mediated TET2 activation and its utility in *TET2$^{MT}$* MN patients we analyzed the nature of the interaction between AA and TET2 catalytic domain. DNA substrate and cofactors α-KG and Fe$^{2+}$ binding to TET2 are essential for its activity, but conditions needed to enable AA-mediated amplification of TET2 activity are poorly understood. Previously, using indirect internal fluorescence quenching assay, it was reported that AA binds to TET2 catalytic domain with an affinity of $8.7 \times 10^3$ M$^{-1}$, however, the binding condition lacked the substrate[22]. Nevertheless, DNA binding is co-operative and may induce structural fitness and enhance the dioxygenase activity[36].

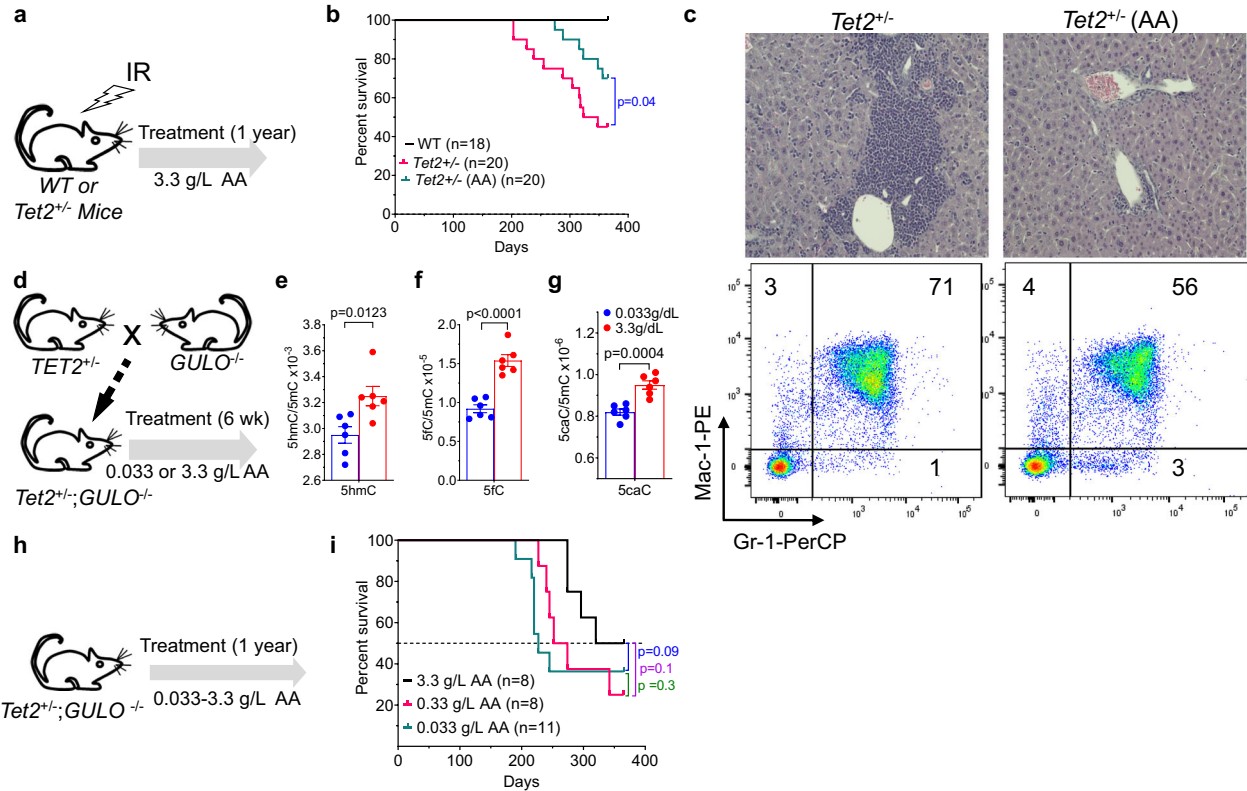

**Fig. 2 Long-term oral AA supplementation prevent the evolution of myeloid neoplasia (MN) in Tet2$^{+/-}$ mice. a** Both WT and *Tet2$^{+/-}$* mice were treated with 4.5 Gy of ionizing radiation. WT mice were maintained on $H_2O$. *Tet2$^{+/-}$* mice were supplied or not supplied with 3.3 g/L AA in $H_2O$. **b** Kaplan–Mayer survival curves for WT and *Tet2$^{+/-}$* mice. AA treatment significantly prolonged survival in irradiated *Tet2$^{+/-}$* mice. **c** Long-term AA treatment of irradiated *Tet2$^{+/-}$* mice reduced liver myeloid cell infiltration (top) and Gr1$^+$/Mac1$^+$ BM cell proportions (bottom). The figures are representative results of 8 mice for each group. **d** *Tet2$^{+/-}$* and *Gulo$^{-/-}$* mice were crossed to create hybrid *Tet2$^{+/-}$;Gulo$^{-/-}$* mice. These mice were given either 0.033 g/L or 3.3 g/L ascorbic acid in their drinking water for six weeks, then sacrificed. **e–g** A 3.3 g/L AA increased DNA oxidation levels in *Tet2$^{+/-}$; Gulo$^{-/-}$* mice. DNA oxidation products (E. 5hmC, F. 5fC, and G. 5caC) were assessed by 2D-UPLC-MS/MS. Data are shown as mean with SEM ($n = 6$). **h** *Tet2$^{+/-}$;Gulo$^{-/-}$* mice were maintained on either 0.033 g/L AA, 0.33 g/L AA, or 3.3 g/L AA in their drinking water for 1 year. **i** 3.3 g/L AA prolonged survival of *Tet2$^{+/-}$;Gulo$^{-/-}$* mice compared to 0.033 or 0.33 g/L AA. Curve comparisons were performed in GraphPad Prizm 8.0.1 and the *p* values are shown for Gehan-Breslow-Wilcoxon test (**b** and **i**) and statistical significance (*p* values) from two tailed *t* test are indicated (**e–g**).

Therefore, to explore the conditions of AA interactions with TET2$^{CD}$ (TET2 catalytic domain) we performed a direct binding assay using surface SPR in the presence of cofactors and methylated cytosine containing DNA substrate. We expressed and purified TET2$^{CD}$ with or without GST tag to homogeneity (Supplementary Fig. 3a) and performed SPR binding analysis with AA. For this purpose, we either immobilized GST-TET2$^{CD}$ on a CM-5 biosensor chip using anti-GST antibody and let AA flow over it, or we immobilized streptavidin and used it to capture biotin labeled 5 mC DNA, followed by binding to TET2$^{CD}$. Different concentration of AA in the presence of 5mC-DNA, 100 μM Fe$^{2+}$ and 10 μM of pseudo substrate NOG were used as analytes, without (1st scenario) or with (2nd scenario) TET2$^{CD}$. GST was used as a reference channel control. Analyses using Biaevalution™ software revealed that AA binds to immobilized GST-TET2 (1st scenario) with dissociation constant K$_d$ = 9.1 ± 2.3 μM in the presence of 5mC-DNA, Fe$^{2+}$ and pseudo substrate NOG (Fig. 4a). In the 2nd scenario, TET2$^{CD}$ without GST tag was captured on 5mC-DNA-biotin immobilized on streptavidin sensor chip and different concentration of AA in the presence of 100 μM Fe$^{2+}$ and 10 μM of pseudo substrate NOG were used as analytes (Fig. 4b). Under these conditions, the dissociation constant was K$_d$ = 11.5 ± 2.2 μM. This value is nearly tenfold greater (stronger binding) than the previously reported K$_d$ for AA binding to TET2, which was estimated via UV-induced protein

internal fluorescence quenching by AA in the absence of 5mC DNA[22]. In the present study, SPR measures interaction in the biologically relevant fully activated complex. In contrast, fluorescence quenching assays are prone to limitations of AA binding to proteins due to its strong absorbance in uv regions (Supplementary Fig. 3b). In addition, AA can quench the fluorescence intensity of disparate proteins such as bovine serum albumin (BSA), Glutathione-S-transferase (GST) and TET2$^{CD}$ with quenching constants of $8.2 \times 10^3$ M$^{-1}$, $18.2 \times 10^3$ M$^{-1}$ and $8.1 \times 10^3$ M$^{-1}$ respectively (Supplementary Fig. 3c–f). These results are consistent with previously reported values for uv-induced fluorescence quenching constant for TET2[22]. This effect in part, may be due to a "sphere of action" quenching mechanism of AA with protein fluorescence[37].

The binding constant of AA ($9.1 \pm 2.3$ μM) with TET2$^{CD}$ calculated from direct binding interaction in SPR was consistent with a half maximal effective concentration of AA (EC50 = 13.8 ± 1.3 μM) for enhancing TET2$^{CD}$ activity as measured by the formation of 5hmC in an ELISA assay (Fig. 4c). Turning to the mechanism of AA enhanced TET2 catalysis, using isolated TET2$^{CD}$ in the presence of α-KG, no effects of AA were observed on TET2 activity in the presence of Fe$^{2+}$ (Fig. 4d, e). The effect of AA was observed only when Fe$^{3+}$ were used. Thus, maintaining the redox state of iron Fe$^{2+}$ in the catalytic site to TET2 is most likely the mechanism of action of AA mediated increase in

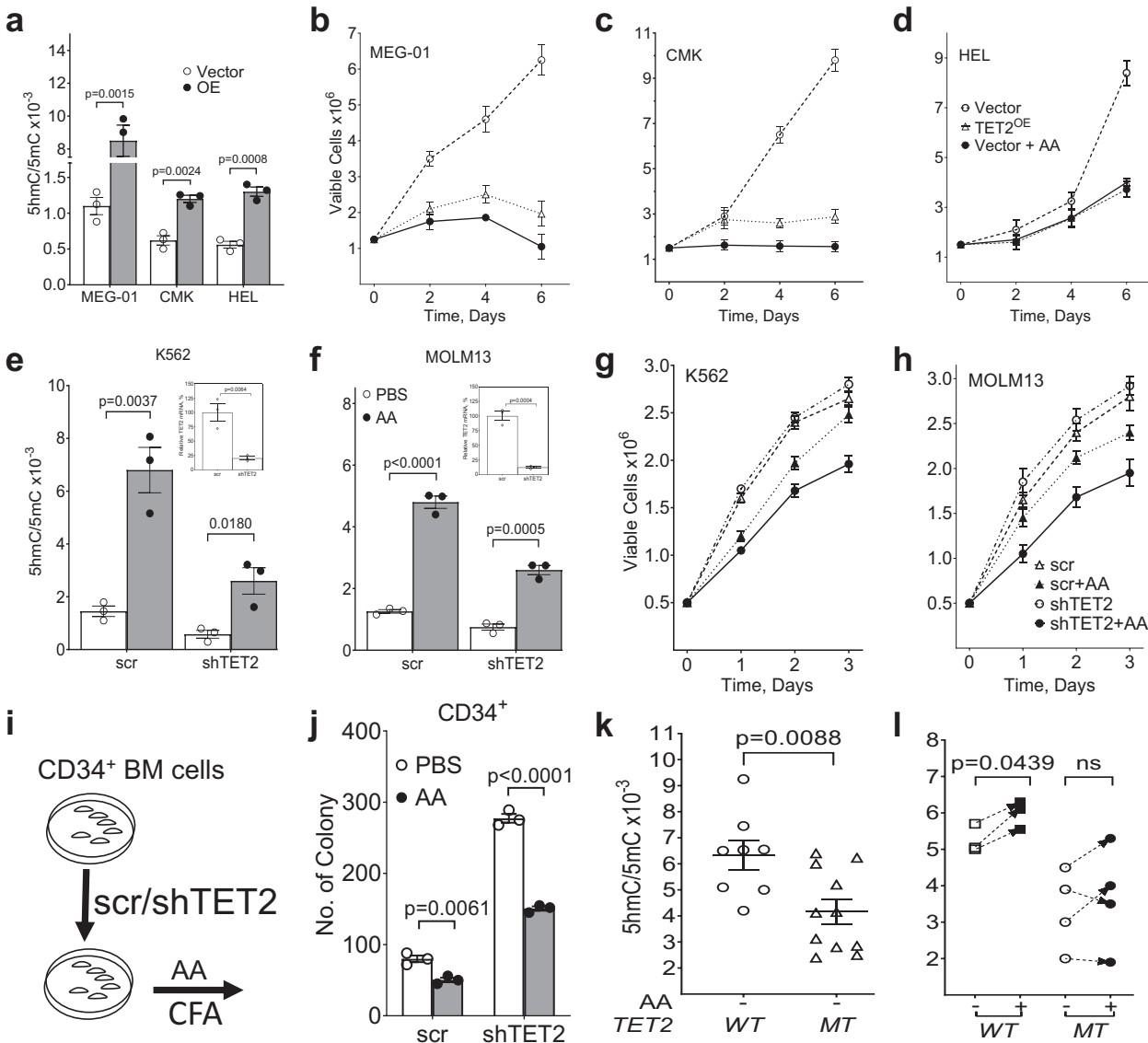

**Fig. 3 Levels of TET2 expression and activity regulates proliferation in human myeloid cells. a** Natural *TET2*-mutant HEL and *TET2* wild-type MEG-01 and CMK cells were stably transfected with either an empty pOZ (Vector) or a pOZ-TET2-overexpression vector (*TET2*^OE^). Levels of 5hmC and 5mC were assessed by dot blot assay. **b–d** Cell lines (Vector and *TET2*^OE^ of HEL, MEG-01, and CMK) were treated for 6 days with either PBS or 250 µM AA. Surviving cells were assessed by trypan blue exclusion assay on Vi-CELL XR cell viability analyzer (Beckman Coulter) and the cell output was plotted. **b** MEG-01. **c** CMK. **d** HEL. **e–h** Human K562 and MOLM13 cells were transduced with scrambled shRNA (scr) or shRNA targeting TET2 (*shTET2*) and treated with 250 µM AA. **e–f** 5hmC and 5mC were assessed after 24 h by 2D-UPLC-MS/MS. **g–h** Cellular proliferation was accessed by the number of viable cells at different time. **i–j** Primary human cord blood CD34+ cells and proliferation was assessed after 14 days by colony forming assay. **a–j** Data are shown as mean ± SEM ($n = 3$) and are representative of two independent experiments. **k** $TET2^{WT}$ ($n = 8$) and $TET2^{MT}$ ($n = 11$) human BM levels of 5hmC/5mC was measured by 2D-UPLC-MS/MS. Data are shown as mean with SEM. **l** $TET2^{WT}$ and $TET2^{MT}$ human BM cells were treated for 48 h with 250 µM AA or left untreated. Base oxidation levels were assessed by 2D-UPLC-MS/MS. Data from the same cell are connected with arrow, $TET2^{WT}$, $n = 3$; $TET2^{MT}$, $n = 4$. $TET2^{MT}$ MN patient information are provided in Supplementary Data 4. Paired *t* test was used in (**l**), *p* values are indicated, ns: not significant.

dioxygenase activity[21]. In such scenario, it is plausible to hypothesize that there can be several other direct or indirect biological factors that may circumvent the effect of AA in compensating loss of TET2 activity that may include the post translational modifications in the catalytic domain.

**Effect of TET2 acetylation on AA activity.** Several post-translational modifications of TET2 have been reported to alter its stability and activity in cancer cell lines, including lysine acetylation[33,34,38]. To see the distribution of most abundant lysine acetylation we ectopically expressed TET2 protein in

HEK293T cells purified by immunoprecipitation and analyzed for lysine modification by LC-MS/MS (Supplementary Fig. 4). Consistent with the earlier reports[33,34], we found K53 and K1478 are the most abundant acetylated lysine residues on N- and C-terminal of TET2 protein respectively (Supplementary Fig. 4c, d).

To test the effects of lysine acetylation on the catalytic activity of TET2 and its impact on the cellular proliferation, we treated different leukemia cell lines with several chemical probe that impact N- and C-terminal lysine acetylation. For this purpose, we used the inhibitors of either class I/II deacetylase or acetyl transferase P300/CBP or the activator of sirtuins the class III deacetylase. Treatment of CMK cells with trichostatin A (TSA), a

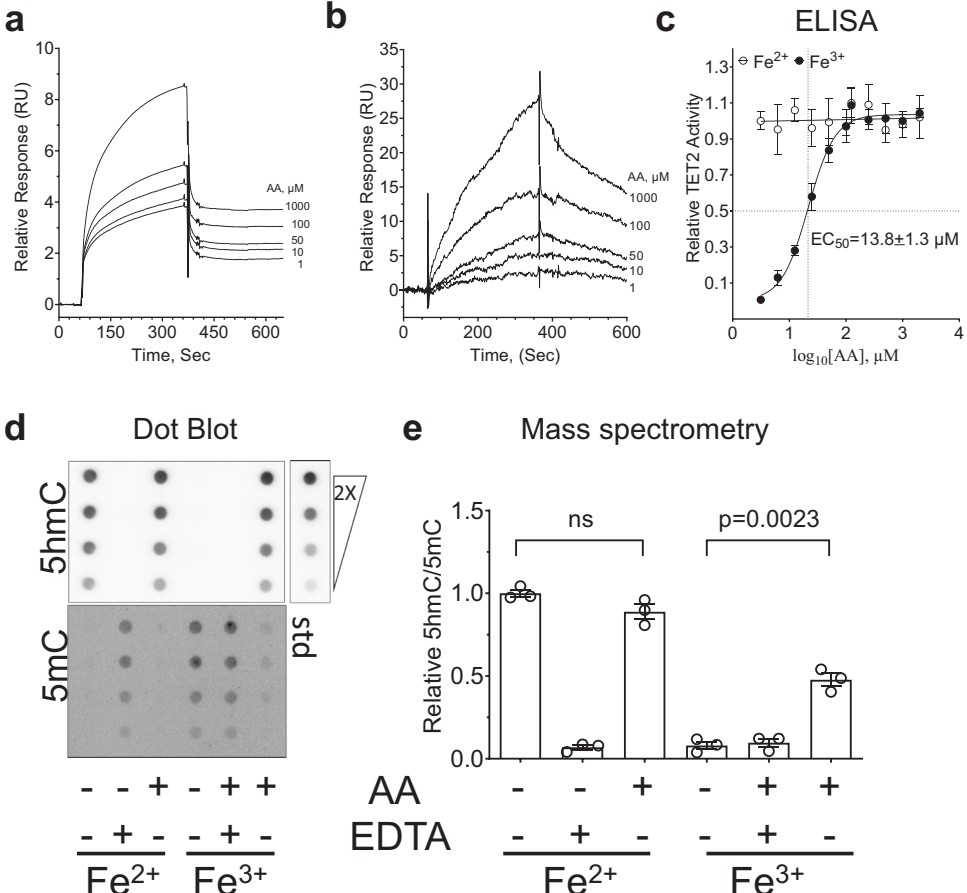

**Fig. 4 Ascorbic Acid binds to TET2 and increases its activity in cell free conditions. a, b** Surface plasmon resonance (SPR) detection of AA [0–1000 μM] binding to purified recombinant GST-TET2$^{CD}$ captured on CM5 biosensor chip immobilized with anti-GST antibody in the presence of a fixed concentration of 5mC-DNA substrate (**b**) or GST tag free TET2$^{CD}$ captured by 5mC-DNA-biotin immobilized on SA chip (**b**), in the presence of NOG (100 μM) and Fe$^{3+}$ (100 μM). The response curve in the absence of AA was used as control and subtracted to generate final response curve. $K_d = k_{off}/k_{on}$. **c.** TET2$^{CD}$ was used in an ELISA based assay with varying concentration of AA to determine the dose response and EC50 = 13.8±1.3 μM was calculated using Log(agonist) vs. response ($R^2 = 0.94$) option in GraphPad Prism 8.0.2. **d–e.** Cell-free DNA combined with purified TET2 (20 nm) and AA (25 μM) for 2 h in the presence of Fe$^{2+}$ or Fe$^{3+}$ (100 μm). **d** dot blot. Top right: 20 pmol 5hmC labeled DNA and its serial twofold dilution were used as positive control. **e** 2D-UPLC-MS/MS. Statistical significance ($p$ values) from two tailed $t$ test are indicated, ns not significant.

known inhibitor of class I/II deacetylase (HDAC I/II)[39], resulted in a dose-dependent increase in TET2 protein level and activity (Fig. 5a, b and Supplementary Figs. 5a-b). This increase in TET2 activity is further enhanced by AA treatment up to threefold (Fig. 5b and Supplementary Fig. 5b). To further expand the understanding of acetylation mediated increase in TET2 activity, we treated *TET2*$^{WT}$ (CMK) and *TET2*$^{MT}$ (HEL and SIG-M5) leukemia cell lines with specific P300/CBP inhibitors C646[40] or HATi[41] and also with the sirtuin activator SRT1720[42] in the presence and absence of AA. The HAT inhibitors or the sirtuin activators both, significantly increased 5hmC/5mC (Fig. 5c and Supplementary Fig. 5c–e). However, treatment with either SRT1720 or HAT inhibitors did not change TET2 protein levels (Supplementary Fig. 5f–g). Treatment of CMK cells with sirtuin activator SRT1720 resulted in a dose-dependent increase in 5 hmC that was further amplified by the cotreatment of AA (Fig. 5c and Supplementary Fig. 5h). While the sirtuin inhibitor, sirtinol[43], demonstrated a dose dependent decrease in TET-activity (Fig. 5d and Supplementary Fig. 5i). To test the growth inhibitory effects of the enhanced TET-activity, leukemia cell lines were treated with sirtuin activator SRT1720 in the present or absence of AA (Fig. 5e–g). SRT1720 treatment on its own had antiproliferative effect with LD50 ranging from 4–8 μM in

leukemia cells (Fig. 5e). There was a ~50% decrease in the LD50 of SRT1720 by the addition of AA (Fig. 5f). AA at concentrations up to 100 μM showed a moderate effect, but increasing SRT1720 induced extensive cell death (Fig. 5g). The increased levels of TDOP along with antiproliferative effect of SRT1720 and AA in SIG-M5, a TET2 null cells suggest that the effects may be due to activation of TET1/3 (Supplementary Figs. 5b and j) and other SIRT1 targets with anticancer activity[44]. To test whether this observation is true for cells derived from MN patients, we isolated BM mononuclear cells from three different cases of *TET2*$^{+/-}$ (Supplementary Data 3) and treated them either with AA alone or in combination with TSA or SRT1720. AA as a single agent had no effect on TET2 activity or anti proliferative effect, however when combined with either TSA or SRT1720, there was a significant increase in 5 hmC/5mC with corresponding growth suppression (Fig. 5h–i, Supplementary Fig. 5k), suggesting the combination of either sirtuin activators or class I/II HDAC inhibitor may potentiate AA effects on MN cells.

**Effect of AA on TET2 lysine mutations in myeloid neoplasia.** We hypothesized that TET2 missense mutation of catalytic domain lysine residues mimicking acetylation may be found in

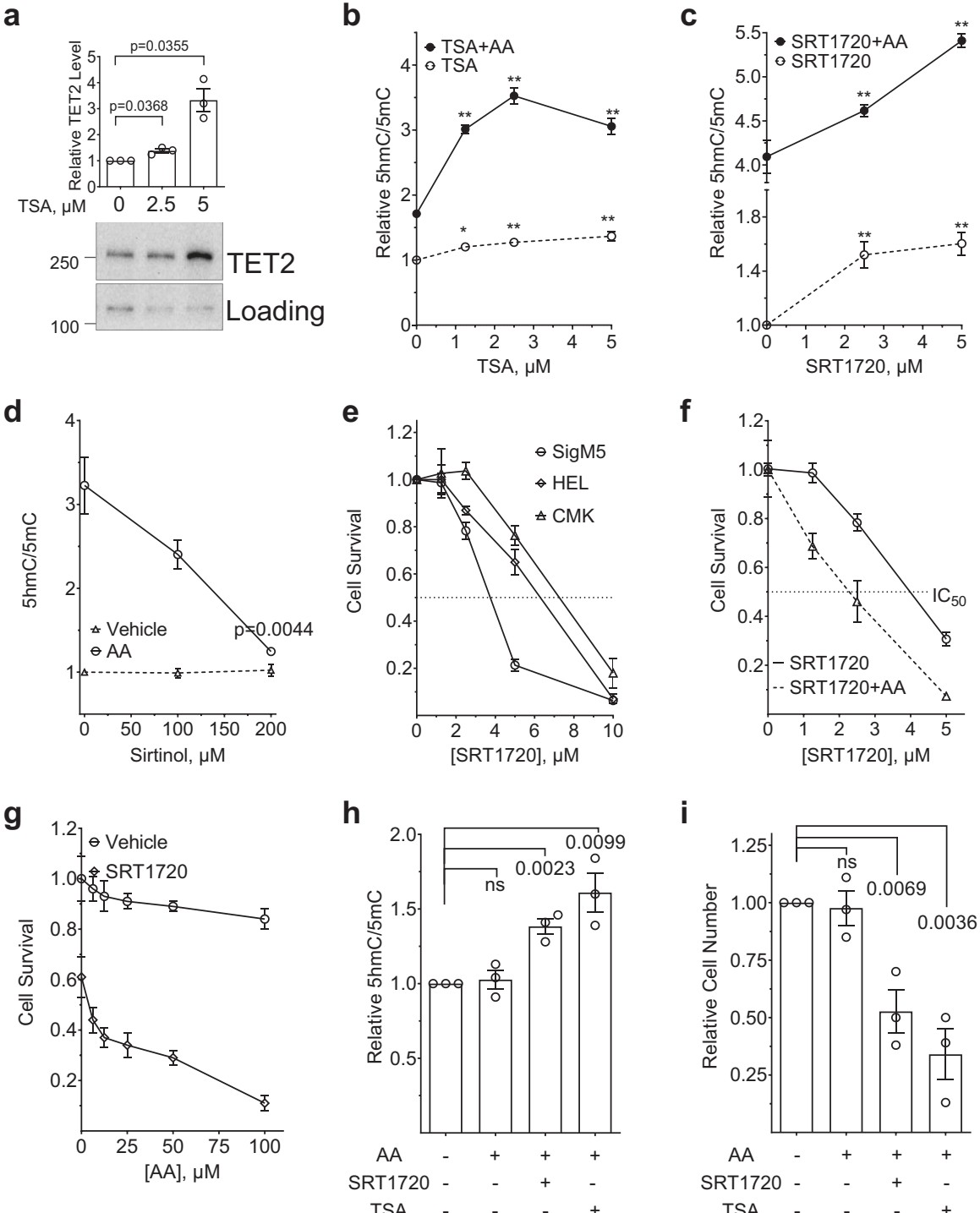

**Fig. 5 Effect of TET2 Acetylation on AA activity. a** CMK cells were treated for 4 h with 0, 2.5 or 5 μM of TSA, nuclear protein fractions were extracted and levels of TET2 protein were detected by western blot. **b** CMK cells were treated for 12 h with 0, 1.25, 2.5 or 5 μM of TSA, in the presence or absence of 100 μM AA. Total genomic DNA was harvested and 5hmC/5mC was determined by dot blot analysis. TSA: Trichostatin A (Cayman, Item No. 89730). **c** CMK cells were treated with increasing concentrations of SRT1720 (Sirtuins activator, Cayman, Catalog #10011020) in the presence or absence of 100 μM AA for 12 h and 5hmC/5mC was determined by dot blot analysis. **d** Sirtinol, a known sirtuin inhibitor prevents AA mediated TET activity. CMK cells were treated with increasing concentrations (0, 100, and 200 μM) of sirtinol (Selleckchem, Catalog #S2804) in the presence or absence of 100 μM AA for 12 h and 5hmC was determined by dot blot analysis. **e** Human myeloid leukemia cell lines CMK, HEL or SIG-M5 with TET2 wild type, heterozygous or homozygous mutation respectively were treated with SRT1720, and cell survival was monitored for 72 h and plotted. **f–g** Sirtuin activation amplifies AA induced TET activity resulting in cell death. CMK cells were treated with increasing concentration of SRT1720 in the presence or absence of 100 μM AA (**f**) or increasing concentration of AA with 2 μM of SRT1720 (**g**) and cell survival was monitored at 72 h post treatment. **h–i** $TET2^{+/-}$ mononuclear cells isolated from MN patient BMs treated with 100 μM of AA or in combination with either 5 μM of TSA or SRT1720 for indicated time. **h** 5hmC/5mC at 24. **i** Cell survival at 72 h. Data are shown as mean ± SEM of triplicate and are representative of at least two independent experiments; statistical significance (p values) from two tailed t test are indicated; *$p < 0.05$, **$p < 0.01$, ns not significant.

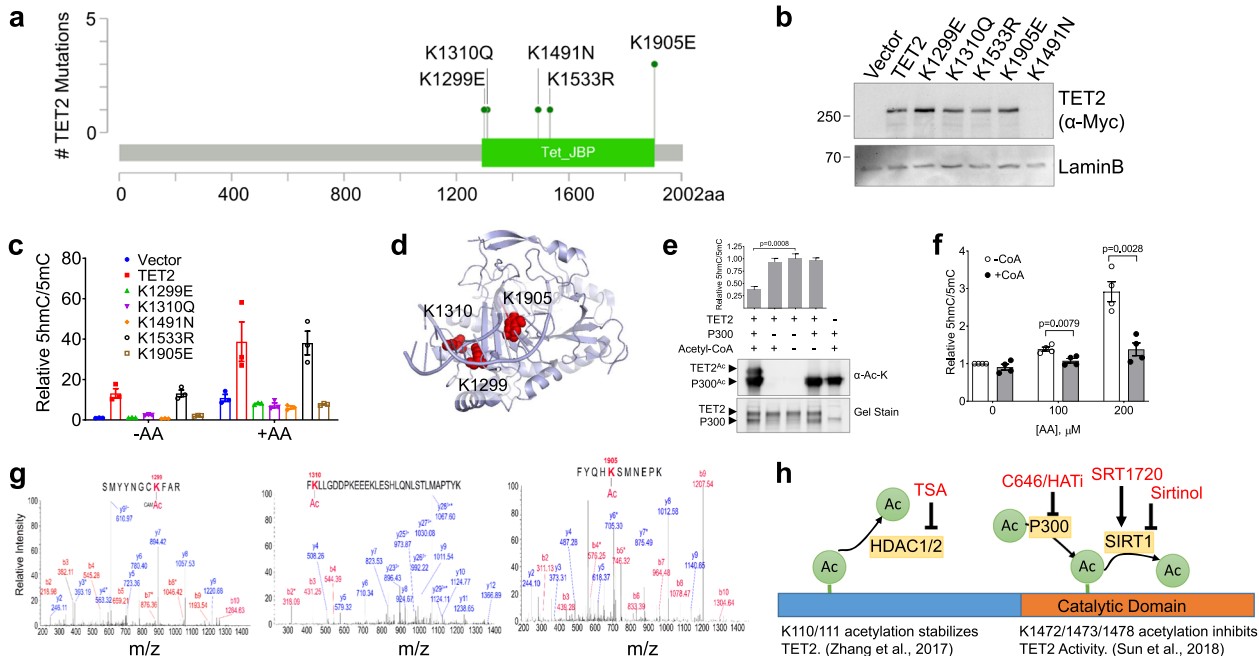

**Fig. 6 Effect of AA on TET2 lysine mutations in myeloid neoplasia. a** Lollipop plot of TET2 missense mutations of lysine residues from a cohort of 4930 patients[1] with Myeloid Neoplasms. **b** Characterization of TET2 lysine mutants identified in patients. The indicated $TET2^{MT}$ were generated by site directed mutagenesis and ectopically expressed in HEK293T cells. Cells were harvested for protein extraction 72 h post-transfection and analyzed by western blot analysis using anti Myc tag antibody (Cell Signaling, Cat# 9402 s, 1:3000). **c** Dot blot for 5hmC/5mC with or without AA (100 μM) treatment for 24 h. **d** Frequently missense mutations in lysine residues of TET2 are present on the DNA binding interface. Cartoon representation of lysine residues K1299, K1310 and K1905 are depicted in red in sphere model. $TET2^{CD}$ (PDB ID 4NM6) was used to generate the figure in Pymol (Deleno Scientific). **e** Lysine acetylation inhibits TET2 activity. *In vitro* acetylation reactions were performed for $TET2^{CD}$ with or without P300/CBP and in the presence or absence of Acetyl-CoA. Anti-acetyl Lysine antibody (Cell Signaling, Cat# 9441, 1:1000) was used for the detection. SDS-PAGE was also stained to visualize the total proteins. Acetylated and nonacetylated $TET2^{CD}$ were used for in vitro activity assay in the presence of 5mC-DNA substrate, 2-OG and $Fe^{2+}$. 5hmC and 5mC levels were assessed by 2D-UPLC-MS/MS. **f** In vitro acetylation reactions were performed for $TET2^{CD}$ with P300/CBP in the presence (+CoA) or absence of Acetyl-CoA (-CoA). Acetylated (+CoA) and nonacetylated (-CoA) $TET2^{CD}$ were used for in vitro activity assay. 100 μM of $Fe^{3+}$ and increasing concentrations (0, 100, and 200 μM) of AA were added in the reaction buffer. 5hmC and 5mC levels were assessed by dot blot. **g** Mass spectrometric analysis was performed for the acetylated $TET2^{CD}$ protein. K1299, K1310, and K1905 were identified as acetylated lysine. **h** Schematics of the regulation of TET2 activity by acetyltransferases and deacetylases. Compounds used to regulate acetyltransferases and deacetylases activity are indicated. Data are shown as mean ± SEM of the triplicate and are representative of two independent set of experiments; statistical significance (*p* values) from two tailed *t* test are indicated (**e**–**f**).

patients with MN, since acetylation leads to loss of TET2 function. TET2 has a total of 136/2002 (6.8%) lysine residues of which 55 are in the catalytic domain (residues 1129–2002). Analysis of the frequency of TET2 lysine residue mutations among 1205 $TET2^{MT}$ patients (24% of 4930 patients)[1] revealed that the majority of them are frameshifts (Supplementary Data 4). Interestingly, all 7 missense mutations (one each of K1299E, K1310Q, K1491N and K1533R and 3 K1905E) were found in the C-terminal catalytic domain (Fig. 6a, Supplementary Data 4). In order to probe the functional consequences of these mutations, we used site directed mutagenesis and generated these missense mutations and ectopically expressed in HEK293T cells. Our data showed that $TET2^{WT}$ and K1299E, K1310Q, K1533R, and K1905E mutant proteins expressed to same levels, with exception of K1491N mutant being very lowly expressed, implicating K1491N mutation may affect the protein stability (Fig. 6b and Supplementary Fig. 6a). The activity analysis using dot blot for 5hmC and 5mC showed that K1299E, K1310Q and K1905E mutants are completely inactive and, interestingly, addition of AA treatment could not restore the loss of TET2 function (Fig. 6c and Supplementary Fig. 6b). Consistent with prediction, lysine to arginine (K1533R) mutation has no effect on TET2 activity (Fig. 6c and Supplementary Fig. 6b), presumably due to the similarity in these two amino acids. Structural analysis of the

$TET2^{CD}$ in complex with 5mC DNA substrate and the cofactors revealed that lysine residues K1299, K1310 and K1905 are at the protein-DNA binding interface (Fig. 6d). Therefore, mutations in TET2 catalytic domain lysine that mimic acetylation, would significantly affect its ability to bind DNA substrate[36] and therefore any mutation or post-translational modification may cause loss of activity which cannot be reversed by amplifying Fe(III)/Fe(II) redox reaction by AA.

To test if the catalytic domain acetylation leads to loss of TET activity that cannot be restored by AA treatment, we performed in vitro acetylation experiment using recombinant $TET2^{CD}$ and P300 in the presence of co-factor Acetyl-CoA and measured enzymatic activity and lysine acetylation by high resolution MS (Fig. 6e–g, Supplementary Fig. 6c). We observed K1299, K1310, and K1905 were acetylated along with previously reported lysines (K1472/1473/1478)[33]. Nearly complete series of b-ions and y-ions were observed and unambiguously assigned to specific indicated peptide sequences. Pairs b or y ions (marked * y3/y4 and b7/b8 for K1299, $y283^+/y293^+$ for K1310, y6/y7 and b4/b5 for K1905) are indicative of acetylation as these fragmentation ions contain a 42 Da mass shift from the natural lysine (Fig. 6g). In addition, the corresponding unmodified peptides for K1310 were also detected. However, unmodified peptide for K1299 and K1905 were not observed probably due to the tryptic digestion at unmodified

lysine residue. The acetylation of catalytic domain lysine residues results in the loss of activity of $TET2^{CD}$ that cannot be reversed by addition of AA (Fig. 6f). Thus, acetylation of the lysine residues in TET2 can have a context dependent opposing effect on its activity (Fig. 6h), for example, N-terminal lysine acetylation by class I and II HDACs prevent ubiquitination, decrease protein degradation, and hence increased activity (Fig. 5a, b). On the contrary, acetylation of catalytic domain lysine residues decrease TET2 activity without impacting the protein level (Supplementary Figs. 5f–g) suggesting a direct involvement of these residues in its activity.

## Discussion

Over the years AA has been extensively investigated as an anticancer agent due to its ability to target most common vulnerabilities of cancer cells presented in redox imbalance, oxygen-sensing and epigenetic reprogramming that include enhancing TET-dioxygenase activity[32,45–47]. Several recent studies have suggested that AA may enhance the activity of αKG-dependent dioxygenases and may thus have therapeutic utility in $TET2^{MT}$ MN[23,26]. However, most of these studies used supra-natural cytotoxic pharmacologic doses of AA and, hence it remains unclear if the effects in $TET2^{MT}$ cases are due to AA mediated partial restoration of TET2-activity or TET2 independent effects of AA on redox activities and oxygen sensing. While the question of the utility of AA in $TET2^{MT}$ MN remains unclear, here by using $Tet2^{+/-}$ mice in combination with radiation as MN disease model, we provide first time evidence of the effect of AA treatment in preventing MN evolution. We show that AA delay the onset of MN and extends survival of $TET2^{+/-}$ and AA deficient strain of $Tet2^{+/-};Gulo^{-/-}$ mice. In these models, which one can view as analogous to human CHIP, AA increases TDOP levels and slow myeloproliferation.

We observed a highly variable effects of AA treatment on TET2 activity depending on the model and the context of TET2 mutations and post-translational modification of TET2 protein. AA treatment increased 2–10 fold of TET activity in different cell lines, as reflected in global levels of TDOP. However, the effect of AA was modest in amplifying TET activity in primary tissue derived from either human or mice. We demonstrate that cell lines with naturally occurring $TET2^{MT}$ are indeed sensitive to AA mediated increase in TET-activity in both heterozygous and biallelic settings. However, the results in a wide variety of primary cells with $TET2^{MT}$ showed a spectrum of responses indicating the presence of modifying factors that may impact the AA effect on TET2 activity. Optimal use of AA in $TET2^{MT}$ MN remains to be determined. We observed that AA increases levels of TDOP in many cell lines, BM samples from $TET2^{MT}$ patients and $Tet2^{mt}$ mice and that this effect is indeed associated with decreased proliferation of TET2-deficient cells. Irrespective of the cell type, TDOP levels correlated with TET2 levels; TET2 overexpression led to slower growth. Similarly, proliferative effects resulting from TET2 deficiency were partially reversed by AA. Interestingly, the effects of AA were also observed in $Tet2^{-/-}$ cells, possibly, in part, due to its effect on Tet1 or Tet3 activity that can also be amplified by AA[26] and well established TET family independent effects of AA on cell growth and survival[47]. In either case, AA likely increases the activity of the all TET enzymes and thus may be effective irrespective of whether $TET2^{MT}$ is hemi- or homozygous or biallelic configuration. These in vitro data imply that AA may have utility as either a low-intensity preventive treatment in $TET2^{MT}$ CHIP or as a high dose treatment of MN[48], consistent with promising in vivo effects of AA in murine models of $TET2^{MT}$ MN[23,26], in particular, $TET2/FLT3-ITD$ induced murine AML model[23].

Earlier reports studying anthocyanidin synthase a 2-oxoglutarate Fe(II)-dependent oxygenase, catalyzes the penultimate step in the biosynthesis of the anthocyanin class of flavonoids can directly interact with AA[49]. The functional relationship of AA to human TET2 activity has been investigated using indirect methods in the absence of DNA as a substrate[22]. Using SPR-based binding analyses we report here that AA binds human TET2 catalytic domain in a dose-dependent manner leading to an increase in dioxygenase activity (EC50 of $13.8 \pm 1.3\ \mu M$) consistent with its binding affinity ($K_d = 9.1 \pm 2.3\ \mu M$) in cell free in vitro conditions. Earlier reports of a nearly tenfold lower affinity, may have been due to the indirect nature of the binding assays and the lack of the activated TET2 complex[22]. Interestingly, the mean AA levels in plasma of healthy controls approach $50\ \mu M$[31], significantly below the earlier reported affinity[22]. Therefore, EC50 of AA for TET2 determined in present study is more consistent and meaningful with the physiologic AA levels in serum[31]. Furthermore, we show that AA may increase $Fe^{3+}$ to $Fe^{2+}$ redox reaction in the catalytic site, since $Fe^{3+}$ by itself fails to activate TET2. AA cannot further amplify the TET2 activity if $Fe^{2+}$ is present in excess. However, when costimulated with AA, $Fe^{3+}$ efficiently restores TET2 dioxygenase activity, strongly indicating that AA can enhance the activity of most $Fe^{2+}$ and αKG -dependent dioxygenases either by reducing the enzyme-bound $Fe^{3+}$ to $Fe^{2+}$ or by acting as an alternate oxygen acceptor during uncoupled decarboxylation cycles. These observations clarify the molecular underpinnings of AA/TET2 interactions and further our understanding of the in vivo activity of AA in $TET2^{MT}$ MN.

Our results suggest AA effects on TET2 activity may be attenuated by several layers of regulations of TET2, explaining potential variability of AA treatment results in different models. For example, phosphorylation of tyrosine or serine/threonine residues of TET2 enhances its stability and activity[48] and acetylation of N- and C-terminal lysines exert opposite effects on TET2 activity[33,34]. N-terminal lysine acetylation increases TET2 protein level by preventing proteasome mediated degradation[30]. Consistent with these reports, cells treated with TSA, a class I and II HDAC inhibitors have increased TET2 protein levels and 5hmC. We show that AA can act synergistically with TSA and further amplifies TET activity. However, C-terminal lysine acetylation inhibits TET2 as reflected in global 5hmC/5mC[33]. Lysine acetylation can prevent the TET2 catalytic cycles due to inefficient recruitment of 5mC-DNA substrate, and therefore, AA fails to activate TET2 in such scenario. Effects of AA may thus depend on context of TET2 lysine acetylation.

TET2 lysine residues K1299, K1310 and K1905 are at the DNA binding interface of $TET2^{CD}$ and are thus important for binding to the 5mC-DNA substrate. Consequently, these residues are susceptible to P300-mediated acetylation and thereby down-modulation of catalytic activity. Conversely, their acetylation can be reversed by sirtuins, a known nicotinamide adenine dinucleotide ($NAD^+$)-dependent deacetylases (class III HDACs). Indeed, we have confirmed that activation of sirtuins by the small molecule SRT1720 increased 5hmC, an effect that is further amplified by AA. In contrast, dose-dependent sirtuin inhibition by sirtinol[43,50] leads to TET2 inhibition as apparent from the loss of 5hmC/5mC. Interestingly, a comprehensive analysis of the configuration of 1205 $TET2^{MT}$ MN patients in a large cohort of MN patient data set[1] revealed that all lysine residues missense mutations (K1299E, K1310Q, K1491N, K1533R, and K1905E) are located on C-terminal of TET2. Functional analysis of these missense mutation suggests that K1299E, K1310Q, and K1905E mutations that may mimic lysine acetylation due to similarity in charge distribution have complete loss of TET2 activity and this loss cannot be reversed by AA treatment. On the other hand, K1533N TET2 mutant is poorly expressed and K1533R had no effect on its activity. This observation has mechanistic and clinical

implications in cases of $TET2^{MT}$ associated disorders including MN. For example, loss of sirtuin deacetylase activity may have significant effect on the acetylation status of catalytic domain lysine residues and thus rendering TET2 less permissive to AA-mediated TET-activation. Moreover, our results suggest new context dependent combination approaches for the reactivating TET2, whereby either HAT inhibitors or sirtuin activators can be combined with AA. Other synergistic combinations, may also include hypomethylating agents azacytidine[51,52] and decitabine[53], which has been reported to upregulate TET2 expression and activity.

In summary, here we propose a complex interplay of various post translational modification of TET2 that have profound effects on its dioxygenase activity. Our results explain the effects of AA on TET2 function and likely other TET enzymes and provide a rationale for the long-term benefit of AA in preventing the evolution of MN.

## Methods

**Patient samples**. Patient BM samples were obtained from healthy controls or patients with MN after informed consent in accordance with Cleveland Clinic IRB-approved protocol. Primary patient BM mononuclear cells were purified by Ficoll (Histopaque®-1077, SIGMA, 10771). Human cord blood was acquired from Cleveland Cord Blood Center, Cleveland, Ohio, and CD34$^+$ cells were isolated by human CD34 MicroBead Kit (Miltenyi Biotec) according to the manufacturer's protocol.

**Cell lines**. Cell lines were purchased within 1 year of their use. Cells were used within 10 passages in fresh culture to prevent any drift. K562 and HEK293T were purchased from ATCC (Manassas, VA) and CMK, MEG-01, MOLM-13, HEL and SIG-M5 were purchased from DMSZ (Braunschweig, Germany). Cells were grown according to guidelines provided with them. Additional details are given in Supplementary Data 2.

**Maintenance and analyses of mice**. Animal care and procedures were conducted in accordance with institutional guidelines approved by the Institutional Animal Care and Use Committee. $Tet2^{-/-}$ mice were generated as previously reported[19]. $Gulo^{-/-}$ mice[35] were used to study AA effects on the progression to MN in $Tet2^{+/-}$ mice. Like humans, $Gulo^{-/-}$ mice need AA from food or drinking water for normal development, therefore, $Gulo^{-/-}$ mice were always maintained with 0.033 g/L AA in drinking water. $Gulo^{-/-}$ mice were crossed with $Tet2^{-/-}$ mice to generate $Tet2^{+/-};Gulo^{-/-}$ mice. After weaning (3 weeks of age), $Tet2^{+/-}$ and $Gulo^{-/-};Tet2^{+/-}$ mice (C57BL/6 background) were divided into three groups each treated with 0.033 g/L (maintenance dose), 0.33 g/L (physiological dose) or 3.3 g/L (treatment dose) of AA in their drinking water. PB was collected by retro-orbital bleeding of mice and was smeared for May-Grunwald-Giemsa staining, and/or subjected to an automated blood count (Hemavet System 950FS). Total white blood cells were obtained after red cell lysis. For histopathology analyses, femurs were fixed in formaldehyde, decalcified, and paraffin embedded. Spleens and livers were treated similarly, omitting the decalcification step. Sections (4.5 μm) were stained with hematoxylin/eosin. For flow cytometric analyses, single-cell suspensions from BM, spleen, and PB were stained with fluorochrome-conjugated antibodies (Gr-1-PerCP, Clone RB6-8C5, BD Biosciences Cat# 552093, 1:200; MAC-1-PE, Clone M1/70, BD Biosciences Cat# 553311, 1:200). Analyses were performed using a BD FACSCantoII or LSRII flow cytometer. All data were analyzed by FlowJo7.6 software.

**TET2 protein purification**. pGEX4T1-TET2$^{CD}$ (1099–1936 Del-insert[54]) expression vector was transformed into *Escherichia coli* strain BL21(DE3)pLysS. The transformant was grown at 37 °C to an OD600 of 0.6 and induced at 16 °C for 18 h. Cells from 2 L culture were harvested and lysed in 50 ml of lysis buffer [20 mM Tris-HCl pH7.6, 150 mM NaCl, 1X CelLytic B (Sigma C8740), 0.2 mg/ml lysozyme, 50 U/ml Benzonase, 2 mM MgCl₂, 1 mM DTT, and 1X protease inhibitor (Thermo Scientific A32965)] for 30 mins on ice. Lysate was sonicated by an ultrasonic processor (Fisher Scientific FB-505 with "½" probe) with an amplitude of 70% for 18 1-min cycles (20 s on and then 40 s off). Lysate was then centrifuged twice at 40,000 × g for 20 min. Supernatant was filtered through the membrane with the pore size of 0.45 μm. Flowthrough was diluted 4 times with the solution of 20 mM Tris-HCl pH7.6, 150 mM NaCl. GST-TET2 was purified by GE Healthcare AKTA pure by affinity (GSTPrep FF16/10) and gel filtration (Superdex 200 increase 10/300 GL). For gel filtration, buffer of 10 mM phosphate and 140 mM NaCl, pH 7.4, was used. GST was removed by TEV protease (Sigma, T4455).

**Fluorescence quenching**. Different concentrations of AA were incubated with 0.4 μg protein in 100 ul HEPES buffer (50 mM HEPES, pH 6.5, 100 mM NaCl,

0.1 mM Fe$^{2+}$, 1 mM DTT and 1 mM αKG). Fluorescence measurements were performed on an F-2500 Fluorescence Spectrophotometer (Hitachi). Proteins were excited at 280 nm wavelength and intrinsic fluorescence emission spectra was measured from 290 to 450 nm. Both excitation and emission slits were 5 nm. The response time was 0.08 s. The binding constant (Ka) was calculated according to the Modified Stern-Volmer equation[22].

**Surface plasmon resonance**. Kinetic characterization of TET2 binding to AA was monitored by surface plasmon resonance (SPR) with a Biacore 3000 (GE Healthcare). Response units (RU), a measure of binding, were monitored as a function of time. To prepare a surface plasmon sensor chip, purified GST tagged TET2$^{CD}$ (purity >90%) was captured by anti-GST antibodies (Biolegend, Cat# 640802, 10 μg/ml)[55,56]. Varying concentrations of AA (0–1000 μM) in the presence of substrate DNA, Fe$^{3+}$ and pseudo substrate NOG (25 μM; replacing the true substrate 2-oxoglutarate, 2-OG) was used as analyte. Alternatively, Biacore chips used immobilized biotin labeled 5mC DNA oligo substrate and GST tag free TET2$^{CD}$ was captured, varying AA across 0–1000 μM in the presence of Fe$^{3+}$ and NOG (25 μM). In all SPR experiments, analyte solutions of different concentrations were passed over the sensor chip containing immobilized protein at a flow rate of 10 μl/min for 5 min, and dissociation was monitored while SPR buffer passed over the chips for an additional 5 min. Data were normalized against a reference channel containing immobilized GST. Surfaces were regenerated using two injections of 500 mM NaCl in HBS-P buffer at 20 μl/min for 30 s. Analysis, and fitting of data, was performed with BIA-Evaluation software, version 3.2 (Biacore Inc.), with the option for simultaneous Ka/Kd calculations. Sensorgram data were fitted using global fits to yield Ka and Kd simultaneously assuming a 1:1 Langmuir model. Goodness-of-fit was accepted based on the criterion of $\chi^2 \leq 1\%$ of the observed maximum response (R$_{max}$).

**Dot blot**. For in vitro reactions, 60 bp duplex DNA substrates (1 μM) were incubated with 0.4 μg TET2$^{CD}$ protein in 100 μl buffer containing 50 mM HEPES (pH 6.5), 100 mM NaCl, 0.1 mM Fe(NH4)₂(SO4)₂ or FeCl₃, AA, and 1 mM alpha-ketoglutarate for 2 h in 37 °C. 60 bp duplex DNA includes: F 5′-ATTACAATATA TATATAATTAATTATAATT AACGAAATTATAATTTATAAT-TAATTAATA-3′ R 5′-TATTAATTAATTATAAATTATAATTT$^m$CGTTAATTATAATTAAT TATATAT-ATATTGTAAT-3′ (IDT Inc) sequences. After reactions duplex DNA was mixed with the same volume of 2X denaturing buffer (0.8 M NaOH/20 mM EDTA) for 10 min at 95 °C and neutralized with equal volumes of 2 M NH4OAc (pH 7.0). Dot blot DNA was extracted using the Genomic DNA Purification Kit (Promega). Samples were denatured and spotted on a nitrocellulose membrane using a Bio-Dot Apparatus Assembly (Bio-Rad). They were then air-dried, cross-linked by Spectrolinker™ XL-1000 (120 mJ/cm²), and detected with anti-5hmC (Active motif, Cat# 39769, 1:5,000) or anti-5mC (Eurogentec, Cat# BI-MECY-0100, 1:2,500) antibodies. Membranes were stained by methylene blue.

**5hmC ELISA for TET2 activity detection**. The 96-well microtiter plate was coated with 10 pmol avidin (0.66 μg, SIGMA A8706) in 0.1 M NaHCO₃. Biotin-5mC-DNA (IDT) substrates were then captured followed by incubation with TET2$^{CD}$ (0.4 μg) in 100 μl assay buffer [50 mM HEPES pH 6.5, 100 mM NaCl, 0.1 mM Fe(NH4)₂(SO4)₂ or FeCl₃, and indicated concentration of AA along with 1 mM 2-OG] for 2 h at 37 °C. Reactions were stopped by adding 0.05 M NaOH (10 μL). After washing, wells were blocked with 2% BSA and probed with anti-5hmC antibody (Active motif, Cat# 39769, 1: 3000) at 4 °C overnight and visualized by HRP-conjugated anti-rabbit secondary antibody (Santa Cruz, Cat# sc-2004, 1:10,000) and developed by TMB (100 μl/well; SIGMA, T4444) and the color development was stopped by adding 50 μl of 2 M H₂SO₄ and measuring optical densities at 450 nm.

**TET2 shRNA transduction**. A $1 \times 10^5$ cells/condition in a total volume of 1 mL were placed in a 1.5 mL Eppendorf tube. 2 μL of Polybrene (4 mg/mL) was added to each tube, followed by 20 μL shTET2 (MISSION® shRNA Lentiviral Transduction Particles, TRCN0000421134, SIGMA) or scrambled shRNA virus. Samples were centrifuged at $1000 \times g$ at 32 °C for 90 min. Supernatants were discarded, samples were resuspended in appropriate media with FBS supplementation, and placed in 6-well plates for further experiments.

**Stable TET2 overexpression**. HEL, MEG-01, and CMK cell lines were stably infected *via* lentivirus with either an empty pOZ vector or a pOZ-TET2 (TET2$^{OE}$) vector. TET2 expression was confirmed by RT-PCR.

**Quantitative real-time PCR**. Total RNA was extracted from cells using RNA purification kits (Macherey-Nagel). High capacity cDNA reverse transcription kits (Applied Biosystems) were used to generate cDNA. CFX96 real-time PCR detection (Bio-Rad) using Taqman gene expression primers TET2 (Hs0032599_m1) and GAPDH (Hs02786624g_m1) was also used. All experiments were duplicated. Gene expression was normalized to GAPDH and compared to controls. For *TET1* and *TET3* mRNA detection, qRT-PCR was performed by using SsoAdvanced Universal

SYBR Green Supermix (Bio-Rad, Cat. 1725270). Primers are listed in Supplementary Data 4.

**Site-directed mutagenesis**. In-Fusion® HD Cloning kit (Takara, 638916) was used for TET2 missense mutation cloning. Primers are listed in Supplementary Data 4. All sequence was confirmed by Sanger sequencing.

**Cellular viability Assay**. Cells were treated with small molecules in the presence or absence of varying concentrations of AA for 0–72 h and assessed using Vi-CELL XR cell viability analyzer (Beckman Coulter).

**In vitro colony-forming assays**. Mononuclear cells derived from BM or purified $CD34^+$ cells treated with shTET2 and empty vector were seeded at 3000 cells per methylcellulose plate (Methocult™, H4435; STEMCELL Technologies). Colonies were scored on day 14.

**In vitro acetylation of TET2$^{CD}$ and enzymatic assay**. Purified TET2$^{CD}$ proteins (4 µg) without GST tag were incubated with or without 2 µg recombinant p300 catalytic domain (p300$^{CD}$, Catalog #BML-SE451-0100, Enzo Life Sciences) in the presence or absence of 100 µM Acetyl-CoA (A2056, Sigma-Aldrich), in 100 µl HAT buffer (50 mM Tris–HCl, pH 8.0, 0.1 mM EDTA, 1 mM DTT) at 30 °C for 1 h. For TET2 enzymatic assay, 10 µl containing 0.4 µg of TET2$^{CD}$ acetylated protein was mixed with 100 µl HEPES buffer (50 mM HEPES, pH 6.5, 100 mM NaCl, 1 mM $CaCl_2$, 1 mM αKG, 1 µM 5mC-DNA substrate), $Fe^{2+}$ or $Fe^{3+}$, and different concentrations of AA. Mock reactions with and without Acetyl CoA or P300$^{CD}$ served as controls. Varying amounts of acetylated protein binding was detected by Biacore™ SPR instrumentation.

**Western blot analysis**. Pellets of 10 million cells treated with or without TSA were resuspended in 250 µl buffer A [10 mM HEPES (pH 7.8), 10 mM KCl, 1.5 mM $MgCl_2$, 0.34 M sucrose, 10% glycerol] and incubated on ice (15 min). NP40 at final concentrations of 0.2% was added to cells and vortexed for 10 s at the highest setting (Vortex genie-2, Scientific Inst). Nuclei collected by centrifugation (5 min, $1300 \times g$, 4 °C) were resuspended in 250 µl buffer B (3 mM EDTA, 0.2 mM EGTA) and lysed on ice (5 min). After centrifuging (5 min, $1700 \times g$, 4 °C) supernatant was collected as nuclear fraction and western blot analyses were performed. All solutions were supplemented with 1X protease inhibitor cocktail (A32965, Thermo). Anti-TET2 antibodies (Bethyl laboratories, Cat# 304-247 A, 1:1000) were used for Western Blot analysis.

**2D-UPLC-MS/MS for modified DNA bases**. All 2D-UPLC–MS/MS analyses were performed according to previously reported methods[57]. Briefly, DNA hydrolysates were spiked with a mixture of internal standards in a volumetric ratio 4:1 to form concentrations of 50 fmols/µL of [$D_3$]-5-hmdC, [$^{13}C_{10}$, $^{15}N_2$]-5-formyl-2'-deoxycytidine (5-fdC), [$^{13}C_{10}$, $^{15}N_2$]-5-carboxyl-2'-deoxycytidine (5-cadC), [$^{13}C_{10}$, $^{15}N_2$]-5-hydroxymethyl-2'-deoxyuridine (5-hmdU) and [$^{15}N_5$]-8-oxodG. Chromatographic separation was performed with a Waters Acquity 2D-UPLC system with photodiode array detector. The first-dimension of chromatography quantified unmodified deoxynucleosides (dN) and 5-methyl-2'-deoxycytidine (5-mdC). The second dimension used a Xevo TQ-S tandem quadrupole MS to quantify all other analytes.

**MS analysis of TET2 acetylation**. The LCMS/MS analysis of TET2 post translational modification were performed as described previously[58]. Briefly, protein bands were excised from the SDS-PAGE gel, washed, destained (50% ethanol, 5% acetic acid) and dehydrated in acetonitrile followed by reduction in DTT and alkylation with iodoacetamide prior to the in-gel digestion by adding 5 µL 10 ng/µL trypsin or chymotrypsin in 50 mM ammonium bicarbonate at room temperature for 18 h. The peptides were extracted (2x) in 30 µL of 50% acetonitrile with 5% formic acid. These two extracts were combined and vacuum dried and resuspended in 30 µl of 1% formic acid for LC-MS analysis. For in solution digestion, acetylated and mock control samples were processed for trypsin digestion and desalted during solid phase extraction. The LC-MS system, ThermoScientific Fusion Lumos mass spectrometer interfaced with dionex Ultimate 3000 UHPLC with Dionex column (15 cm × 75 µm id Acclaim Pepmap C18, 2 µm, 100 Å reversed- phase capillary chromatography column) were used for data collection. The extracts from trypsin digest (5 µL) were injected and the peptides eluted using acetonitrile and 0.1% formic acid gradient at a flow rate of 0.3 µL/min. The microelectrospray ion source is operated at 1.9 kV.

**Mass spectroscopic data analysis**. Analysis of MS raw files were performed by Proteome Discoverer. MS/MS spectra were searched against TET2 (NCBI Reference Sequence: NP_001120680.1) or an in-house constructed Human TET2 protein sequence. Modifications considered included acetylation of Lysine (K). Precursor tolerance was set to 10 ppm and the fragment tolerance was set to 0.02 Da. MS-MS Spectra assigned to acetylation were manually inspected to assure

correctness: y- and b- series fragmentation ions were labeled for inspection. Label-free Quantification (total ion intensity) of peptides were performed using Qual Browser in Thermo Xcalibur software. Briefly, peptide retention time and peptide masses were obtained from the associated MS-MS spectra. Isotopically resolved precursor ion m/z ratios were extracted from the LC-MS and used for the selection by column retention time of interest. The area under the peak was used to quantify particular series of ion abundance.

**Statistics and reproducibility**. All statistical analyses were performed in Graph-Pad Prism 8.0 (https://www.graphpad.com/) unless otherwise described. The statistical significance were performed using two tailed student $t$ test unless described otherwise. Each experiment was performed in triplicate at least twice wherever possible.

**Reporting summary**. Further information on research design is available in the Nature Research Reporting Summary linked to this article.

## Data availability

All source data reported in this manuscript is the part of supplementary data as Supplementary Data 1. Any specific material request can be made to the corresponding authors. All engineered cell lines used in this manuscript are available upon request. The mass spectrometry proteomics data for lysine acetylation of TET2 post translational modifications have been deposited to the ProteomeXchange consortium via the PRIDE[59] partner repository with the dataset identifier PXD020550.

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

## Acknowledgements

We appreciate the technical assistances from Proteomics and Metabolomics Core (mass spectrometry), Molecular Biotechnology Core (Surface Plasmon Resonance), and Dennis J. Stuehr (fluorescence experiments), Lerner Research Institute, Cleveland Clinic and animal tumor core taussig cancer center. This work was supported in parts by grants from the NIH (R35HL135795 and RO1 HL132071 to J.P.M.; HL145883 to MX), Leukemia and Lymphoma Society (6582-20-LLS to BKJ, M1701632 to MX), MDS Research Taub Foundation (J.P.M. and B.K.J.), Edward P. Evans Foundation, 2015 Basic Research Grant, American Cancer Society Research Scholar Grant 123436-RSG-12-159-01-DMC, Department of Veterans Affairs BX001820, Polish National Science Centre grants number: DEC-2015/19/B/NZ5/02208, and DEC-2017/27/B/NZ7/01487. The Fusion Lumos instrument for LCMS/MS was purchased via an NIH shared instrument grant, 1S10OD023436-01.

## Author contributions

Y.G. designed research studies, acquired data, analyzed data, and write the manuscript. E.F.G. acquired data, analyzed data, and write the manuscript. M.H., C.M.K., T.R., V.V., H.M. conducted experiments, acquired and analyzed data, provided reagents and assisted in writing relevant sections of the manuscript. S.C., X.L. conducted in vivo experiments, acquired and analyzed data. X.G., B.W. analyzed the proteomic mass spectrometry data and helped writing the relevant results section in the manuscript. D.G., E.Z., and R.O. performed experiments, acquired and analyzed data and helped writing the relevant sections in the manuscript. A.N., M.A.S., Y.S., M.M. provided reagents, and assisted in writing relevant sections of the manuscript discussed the results. M.X. designed and supervised research studies, acquired data, analyzed data, provided reagents, and assisted in editing relevant sections of the manuscript. J.P.M. conceived and conceptualize the idea, designed and supervised the research studies, edited the manuscript. B.K.J. conceived and conceptualize the idea, designed and supervised the research studies, acquired and analyzed data, develop the reagents, wrote and edited the manuscript.

## Competing interests

The authors declare no competing interests.
