## [Peer Review File · Communications Biology]

Reviewers' comments:

Reviewer #1 (Remarks to the Author):

Using murine models and in vitro studies with human cell lines and primary cells, Drs. Guan, Greenberg and colleagues study the effect of ascorbic acid (AA) on TET2 activity and function. While several (high-profile) studies have been published, demonstrating AA enhancement of TET2 activity and dampening of leukemogenic properties in TET2-mutant cells and models, their is perhaps the first to examine a 'direct genetic model of Tet2 deficiency' and to consider the post-translational regulation of TET2 lysine acetylation and its effects on AA response. Their study points to factors that enhance or attenuate AA effects on TET2 activity and may begin to clarify more rationale therapeutic approaches, including combination therapies for patients with myeloid neoplasms.

Major comments:

1. The authors acknowledge that, "In contrast to the genetically engineered mouse cells and human leukemic cell lines, AA only modestly increased TDOP levels in TET2WT while TET2MT showed a [sic] mixed results" in primary cells from myeloid neoplasm patients. In a very interesting set of experiments that follow with recombinant TET2-CD and cell lines, they then show that post-translational acetylation of lysine residues within the catalytic region of TET2 modulates the AA effect on TET2 activity. They also mention "several layers of TET2 activity regulation" that include phosphorylation of tyrosine or serine/threonine residues, along with opposing effects of N- and C-terminal lysine acetylation. However, they never come back to the variable results with primary human BM cells from MN patients in Figures 3 K-L, and close the loop by potentially attributing the variability to the factors identified in the remainder of the paper.

2. The authors observe some effects of AA in Tet2^{-/-} cells and mention it could be "possibly due to Tet1 or Tet3 activity that can also be amplified by AA" (line 265). This, however, is entirely speculative unless they cite or provide evidence of Tet1 or Tet3 expression in the particular cell types examined. On this note, especially in the AA treatment models of Tet2^{+/-} mice (long-term model, ionizing radiation cohort, Tet2^{+/-}-Gulo^{-/-} cross), the authors should at least discuss possible Tet2-independent (or Tet family-independent) effects of AA. They have not run parallel Tet2 homozygous controls.

Minor comments:

1. The authors should consider re-writing the abstract to provide more context and definitions. For example, for a reader who may not be familiar with the background literature, the relationship between AA and TET2 may not be clear.

Also, what is "preventative mode"? More specifics could be provided - what is meant by "patient derived cells" and "specific chemical probes". "MDS/MPN" is mentioned but not defined in concluding sentence. Only "myeloid neoplasia/MN" appears before this. "Sirtuin" is only mentioned in conclusion.

2. The sentence on line 57 and the one that spans lines 59-60 are essentially the same and are redundant.

3. Line 63 - a more appropriate/specific term than "recharging" should be used.

4. Lines 88-89 (and throughout): please carefully check the grammar. "In vivo effects of AA in preventing leukemogenesis was [sic] established in in [sic] transplant models..."

5. Line 92 - please provide more specifics about the types of cells used, rather than just "mouse BM

and spleen cells". These compartments obviously contain a variety of cells.

6. Line 101 - the description of the results is inaccurate. The authors write: "indicated by lower spleen size, total WBC, and myeloid lineage counts" when, in fact, they show in the figure lower spleen WEIGHT, MONOCYTE counts and myeloid lineage PROPORTIONS".

7. Line 105 and Fig. 2 legend - were WT mice also exposed to sublethal ionizing radiation, as stated? This is not clear.

8. Line 109 and Fig. 2C - it is stated that the reduction is 20% but this is not shown in the figure. Also, is this a representative result? Finally, the histological image is too small and doesn't show convincing evidence of myeloid infiltration, based on morphology, at this magnification.

9. Lines 114-115 - the description is vague and it should be qualified that the results do not achieve statistical significance.

10. Lines 121-122 - a description of TET2 mutation status/zygosity appears later in the manuscript. It would be helpful to provide some of this context for the cell lines earlier on and to justify the various choices of cell lines.

11. Lines 131-133 - are these statistically significant results?

12. Line 137 - BM samples from MN patients - please provide more specifics. What samples, how isolated or fractionated?

13. line 226 - perhaps provide a name for the "inhibitor of sirtuin".

14. lines 235, 238-239 - the "profound anti-proliferative effect" of sirtuin activation on its own requires some explanation/discussion, particularly the "extensive cell death" in homozygous TET2mt cells.

15. line 250 - the claim that "direct evidence" is provided in the murine model requires further explanation/justification. It could be argued that the authors have not excluded TET2-independent effects.

16. lines 268-269 - what is "higher density treatment"?

17. Figure 1 legend - the concentration(s) of AA should be mentioned in the legend, along with numbers of animals used.

18. Figure 2 legend - please include numbers used (n=) where not indicated. Where WT mice also exposed to IR?

19. Figure 3 and legend - is the AA treatment in the figure only for TET2-OE cells? What about parental? What about n= again in this fig? MOLM cells are shown in the figure but not mentioned in the legend. For Fig. 3L it is stated in the legend that n=3 for WT but the plot seems to only show n=1?

20. line 553 - sentence is cut off: "and levels of TET2..."

21. Supplemental figure legend S1- More detail please - # cells, methods for detecting in B and C. Label "F" is missing ahead of sentence about 14 day colony assay? No stats for part G?

22. Figure S4 legend - "presence or absence of 00 uM AA" (00???)

23. Nuclear fractions for western blots - this reviewer does not recall where these were used. Please remind or clarify where used.

Reviewer #3 (Remarks to the Author):

Guan and colleagues investigated the function of AA on TET2 mutant myeloid neoplasia. The authors used animal models, in vitro culture systems and biochemical analysis to demonstrate the impact of AA on Tet2. The authors employed mass spectrometry and pharmacological tools to demonstrate the effect of TET2 acetylation on AA activity. While the experiments and data are convincing, it is unclear how the author could directly connect the various acetylation level of TET2 to AA induced growth-suppressive effect in the mouse models. The molecular mechanism is not very well justified. Additional experiments are required.

1. Figure 2D-I, Tet2^{-/-}-GULO^{-/-} without AA treatment need to be shown as control.
2. Figure 3G, the labels for y-axis and x-axis are missing.
3. The author observed increased protein level as well as the catalytic activity in TSA treated condition. It is unclear whether the increased catalytic activity is due to increased protein level or the acetylation modification could directly enhance the Tet2 catalytic activity.
4. Since TSA could enhance the TET2 protein level, it is unclear whether the inhibitor used in the following study (HATi, C646, SRT1720) could also affect the TET protein level. More appropriate controls are needed to clarify this point.
5. Although the authors observed enhanced catalytic activity of TET2 and growth suppressive effect using pharmacology inhibitors to enhance the acetylation modification of TET2 in leukemia cells, it is unclear whether these effects are directly due to the increased acetylation on the residues mentioned in the manuscript. A further MS analysis and mutational analysis could strengthen this conclusion.

Response to the Reviewers' comments:

Reviewer #1 (Remarks to the Author):

Using murine models and in vitro studies with human cell lines and primary cells, Drs. Guan, Greenberg and colleagues study the effect of ascorbic acid (AA) on TET2 activity and function. While several (high-profile) studies have been published, demonstrating AA enhancement of TET2 activity and dampening of leukemogenic properties in TET2-mutant cells and models, this is perhaps the first to examine a 'direct genetic model of Tet2 deficiency' and to consider the post-translational regulation of TET2 lysine acetylation and its effects on AA response. Their study points to factors that enhance or attenuate AA effects on TET2 activity and may begin to clarify more rationale therapeutic approaches, including combination therapies for patients with myeloid neoplasms.

We are thankful for the insightful comments and the recognition of our work.

Major comments:

1. The authors acknowledge that, "In contrast to the genetically engineered mouse cells and human leukemic cell lines, AA only modestly increased TDOP levels in TET2WT while TET2MT showed a [sic] mixed results" in primary cells from myeloid neoplasm patients. In a very interesting set of experiments that follow with recombinant TET2-CD and cell lines, they then show that post-translational acetylation of lysine residues within the catalytic region of TET2 modulates the AA effect on TET2 activity. They also mention "several layers of TET2 activity regulation" that include phosphorylation of tyrosine or serine/threonine residues, along with opposing effects of N- and C-terminal lysine acetylation. However, they never come back to the variable results with primary human BM cells from MN patients in Figures 3 K-L, and close the loop by potentially attributing the variability to the factors identified in the remainder of the paper.

Thank you for pointing out the gap in our discussion and the presentation of overall idea. We agree and revised the manuscript accordingly in the light of new data generated utilizing human bone marrow samples, mutagenesis, LCMS/MS and TET2 activity data. We have now compared and contrasted AA effect in different model systems and closed the loop of our findings by examining the effect of TSA and SRT1720 in primary patient bone marrow derived mononuclear cells.

This part has been summarized in both result and discussion sections.

Lines 226-231:

"To test whether this observation is true for cells derived from MN patients, we isolated BM mononuclear cells from three different cases of *TET2*^{+/-} (**Table S2**) and treated them either with AA alone or in combination with TSA or SRT1720. AA as a single agent had no effect on TET2 activity or anti proliferative effect, however when combined with either TSA or SRT1720 there was a significant increase in 5hmC/5mC with corresponding growth suppression (**Figs. 5H-I, S5J**)."

Line 352-357:

"This observation has mechanistic and clinical implications in cases of *TET2*^{MT} associated disorders including MN. For example, loss of sirtuin deacetylase activity may have significant effect on the acetylation status of catalytic domain lysine residues and thus rendering TET2 less permissive to AA-mediated TET-activation. Moreover, our results suggest new context dependent combination approaches for the reactivating TET2, whereby either HAT inhibitors or sirtuin activators can be combined with AA."

2. The authors observe some effects of AA in Tet2^{-/-} cells and mention it could be "possibly due to Tet1 or Tet3 activity that can also be amplified by AA" (line 265). This, however, is entirely speculative unless they cite or provide evidence of Tet1 or Tet3 expression in the particular cell types examined. On this

note, especially in the AA treatment models of Tet2+/- mice (long-term model, ionizing radiation cohort, Tet2+/-Gulo-/- cross), the authors should at least discuss possible Tet2-independent (or Tet family-independent) effects of AA. They have not run parallel Tet2 homozygous controls.

Thanks to the reviewer for pointing out the lack of referencing for the TET1 and TET3 in backing our statement. This has been modified in the revised text. We performed qRT-PCR to detect *TET1* and *TET3* mRNA levels for the *TET2*^{-/-} cell line, SIGM5. Our data showed that both *TET1* and *TET3* are expressed in *TET2*^{-/-} SIGM5 cells and the AA treatment can increase the TDOPs in these cells. These results are presented in **Fig. S5B** and **S5G**. We also included a reference paper for Tet1 and Tet3 expression in Tet2^{-/-} (**PMID: 28823558**). This reference paper showed *Tet3* and *Tet2* expression are similarly expressed while *Tet1* expressing is far less. New result and discussion have been included.

Lines 224-226:

“The increased levels of TDOP along with anti-proliferative effect of SRT1720 and AA in SIG-M5 a TET2 null cells suggest that the effects may be due to activation of TET1/3 (Figs. S5B and G) and other SIRT1 targets with anticancer activity⁴⁴.”

Lines 300-304:

“Interestingly, the effects of AA were also observed in *Tet2*^{-/-} cells, possibly, in part, due to its effect on Tet1 or Tet3 activity that can also be amplified by AA²⁶ and well established TET family independent effects of AA on cell growth and survival⁴⁷. In either case, AA likely increases the activity of the all TET enzymes and thus may be effective irrespective of whether *TET2*^{MT} is hemi- or homozygous or biallelic configuration.”

We agree with the reviewer that AA effect in cancer can come independent of its ability to activate TET function. For example, the anticancer property of ascorbic acid may be due to its ability to target most common vulnerabilities of cancer cells presented in redox imbalance, oxygen-sensing and epigenetic reprogramming that include enhancing TET activity (PMID: 30967651). This discussion has been added.

Lines 274-277:

“Over the years AA has been extensively investigated as an anti-cancer agent due to its ability to target most common vulnerabilities of cancer cells presented in redox imbalance, oxygen-sensing and epigenetic reprogramming that include enhancing TET-dioxygenase activity^{32, 45-47}.”

Our recent study of myeloid neoplasia patient with *TET2* mutations showed that the prevalence of *TET2*^{MT} increased with patient age, and it represent phenotype-neutral ubiquitous ancestral hits, rather than leukemic drive (PMID: 29795413). In mice, approximately half of *Tet2*^{-/-} mice die before 1 year old (PMID: 28440315), while *Tet2*^{+/-} mice do not develop overt myeloid neoplasia until 1 year of age (PMID: 21803851). Therefore, *Tet2*^{+/-} better suited to test the TET2 dependent effect of AA.

Minor comments:

1. The authors should consider re-writing the abstract to provide more context and definitions. For example, for a reader who may not be familiar with the background literature, the relationship between AA and TET2 may not be clear.

Also, what is "preventative mode"? More specifics could be provided - what is meant by "patient derived cells" and "specific chemical probes". "MDS/MPN" is mentioned but not defined in concluding sentence. Only "myeloid neoplasia/MN" appears before this. "Sirtuin" is only mentioned in conclusion.

Thank you for the suggestion. We have modified the abstract for clarity that emphasize our findings.

“Loss-of-function TET2 mutations (*TET2*^{MT}) are common in myeloid neoplasia. TET2, a DNA dioxygenase, requires 2-oxoglutarate and Fe(II) to oxidize 5-methylcytosine. *TET2*^{MT} thus result in hypermethylation and transcriptional repression. Ascorbic acid (AA) increases dioxygenase activity by

facilitating Fe(III)/Fe(II) redox reaction and may alleviate some biological consequences of $TET2^{MT}$ by restoring dioxygenase activity. Here, we report the utility of AA in the prevention of $TET2^{MT}$ MN, clarify the mechanistic underpinning of the TET2-AA interactions, and demonstrate that the ability of AA to restore TET2 activity in cells depends on N- and C-terminal lysine acetylation and $TET2^{MT}$. Consequently, pharmacologic modulation of acetyltransferases and histone deacetylases may regulate TET dioxygenase-dependent AA effects. Thus, our study highlights the contribution of factors that may enhance or attenuate AA effects on TET2 and provides a rationale for novel therapeutic approaches including combinations of AA with class I/II HDAC inhibitor or sirtuin activators in $TET2^{MT}$ leukemia."

2. The sentence on line 57 and the one that spans lines 59-60 are essentially the same and are redundant.

Thank you for pointing out the redundancy of sentences. We have modified the segment as per the suggestion.

3. Line 63 - a more appropriate/specific term than "recharging" should be used.

Thanks for your comments. we have changed recharging to more appropriate scientific term "reducing" the new sentence read: Ascorbic acid (AA) enhances the activity of TET2 likely by reducing catalytic site Fe(III) to Fe(II) thus restoring catalytic activity (lines 61-62).

4. Lines 88-89 (and throughout): please carefully check the grammar. "In vivo effects of AA in preventing leukemogenesis was [sic] established in in [sic] transplant models..."

Thank you for pointing out our mistakes. The whole paragraph (lines 89-105) has been corrected.

5. Line 92 - please provide more specifics about the types of cells used, rather than just "mouse BM and spleen cells". These compartments obviously contain a variety of cells.

Thanks for your comments. We have provided the specifics about these cells used in the study.

"mouse BM and spleen cells" has been changed to "mononuclear cells purified from mouse BMs and spleens".

6. Line 101 - the description of the results is inaccurate. The authors write: "indicated by lower spleen size, total WBC, and myeloid lineage counts" when, in fact, they show in the figure lower spleen WEIGHT, MONOCYTE counts and myeloid lineage PROPORTIONS".

Thanks for your comments to make our statement more accurate. The sentence has been corrected.

7. Line 105 and Fig. 2 legend - were WT mice also exposed to sublethal ionizing radiation, as stated? This is not clear.

Thank you for pointing out this confusion. WT mice were also exposed to sublethal ionizing radiation. We have modified the relevant segments in the revised manuscript.

Line 106: We also exposed both cohorts of WT and $Tet2^{+/-}$ mice to sublethal ionizing radiation (IR).

Line 519:(Figure legend): Both WT and $Tet2^{+/-}$ mice were treated with 4.5 Gy of ionizing radiation.

8. Line 109 and Fig. 2C - it is stated that the reduction is 20% but this is not shown in the figure. Also, is this a representative result? Finally, the histological image is too small and doesn't show convincing evidence of myeloid infiltration, based on morphology, at this magnification.

Thank you for pointing out this confusion. Yes. Fig. 2C is a representative result of 8 mice. Fig. 2C has been corrected with better resolution figures. We regret the loss of resolution due to file conversion. The figure legend and manuscript text has been modified to reflect this aspect.

Lines 522-524:

“(Figure legend): C. Long-term AA treatment of irradiated *Tet2*^{+/-} mice reduced liver myeloid cell infiltration (top) and Gr1⁺/Mac1⁺ BM cell proportions (bottom). The figures are representative results of 8 mice for each group.”

Lines 109-111 (result): Oral AA treatment improved the 1-year OS of IR exposed *Tet2*^{+/-} mice to 70%, and reduced BM myeloid-lineage cells fraction by approximately 20% in average, compared to the untreated IR-*Tet2*^{+/-} mice (**Fig. 2C**).

9. Lines 114-115 - the description is vague and it should be qualified that the results do not achieve statistical significance.

Thanks for your suggestions, we modified the sentence in the revised manuscript.

Lines 117-121:

“We observed a dose dependent increase in the survival of *Tet2*^{+/-}*Gulo*^{-/-} mice upon AA treatment. The median survival increased 36 days for 0.33 g/L and 116 days for 3.33 g/L AA supplementation compared to the minimal dose of AA supplementation (0.033g/L) required for the sustenance of these mice, however, this effect does not achieve statistical significance in our experimental settings (**Fig. 2H-I**).”

10. Lines 121-122 - a description of TET2 mutation status/zygosity appears later in the manuscript. It would be helpful to provide some of this context for the cell lines earlier on and to justify the various choices of cell lines.

Thanks for your suggestion. Following changes have been made.

Lines 126-127:

“For this purpose, we ectopically expressed TET2 into both *TET2*^{WT} (MEG-01 and CMK) and *TET2*^{MT} (HEL) leukemia cell lines (**Table S1**).”

11. Lines 131-133 - are these statistically significant results?

As per the reviewer’s suggestion we reanalyzed the data for statistical significance and presented in Figs. **S2D-E**. The text has been modified accordingly and presented.

Lines 135-137:

“AA treatment significantly reduced the growth of *TET2*^{KD} cells as well as the parental cells. However, the AA impact on the growth of *TET2*^{KD} cells was more pronounced (**Figs. 3E-H and S2D-E**).”

12. Line 137 - BM samples from MN patients - please provide more specifics. What samples, how isolated or fractionated?

We have provided the detailed information.

Lines 141-143:

“We also studied AA-mediated up modulation of TET2 activity in primary bone marrow derived mononuclear cells isolated from MN patients with *TET2*^{WT} or *TET2*^{MT} (**Table. S2**).”

Line 549:

TET2^{MT} MN patient information are provided in **Table S2**.

Following sentence has been added in method: Primary patient bone marrow mononuclear cells were purified by Ficoll (Histopaque®-1077, SIGMA, 10771).

13. line 226 - perhaps provide a name for the "inhibitor of sirtuin".

We provided the name of the sirtuin inhibitor and modified the sentence accordingly.

Lines 217-218:

While the sirtuin inhibitor, sirtinol, demonstrated a dose dependent decrease in TET-activity (**Figs. 5D and 5I**).

14. lines 235, 238-239 - the "profound anti-proliferative effect" of sirtuin activation on its own requires some explanation/discussion, particularly the "extensive cell death" in homozygous TET2mt cells.

Thanks for the suggestion. "profound anti-proliferative effect" and "extensive cell death" in homozygous TET2mt cells has been discussed. We also corrected the sentence to reflect our observation.

Lines 220-226:

"SRT1720 treatment on its own had anti-proliferative effect with LD50 ranging from 4-8 μ M in leukemia cells (**Fig. 5E**). There was a ~50% decrease in the LD50 of SRT1720 by the addition of AA (**Fig. 5F**). AA at concentrations up to 100 μ M showed a moderate effect, but increasing SRT1720 dose induced extensive cell death (**Fig. 5G**). The increased levels of TDOP along with anti-proliferative effect of SRT1720 and AA in SIG-M5 a TET2 null cells suggest that the effects may be due to activation of TET1/3 (**Figs. 5B and G**) and other SIRT1 targets with anticancer activity⁴⁴."

15. line 250 - the claim that "direct evidence" is provided in the murine model requires further explanation/justification. It could be argued that the authors have not excluded TET2-independent effects.

Thanks for the reviewer for insightful comment. We have modified the sentence to reflect our observation.

Lines 281-284:

"While the question of the utility of AA in *TET2^{MT}* MN remains unclear, here by using *Tet2^{+/-}* mice in combination with radiation as MN disease model, we provide first time evidence of the effect of AA treatment in preventing MN evolution."

16. lines 268-269 - what is "higher density treatment"?

The sentence was modified to reflect the high or low dose treatment in therapeutic settings.

"Higher density treatment" has been changed to "high dose treatment".

17. Figure 1 legend - the concentration(s) of AA should be mentioned in the legend, along with numbers of animals used.

Thanks for your suggestion. AA concentration has been mentioned in the legend. Numbers of animals used are added in figure.

18. Figure 2 legend - please include numbers used (n=) where not indicated. Where WT mice also exposed to IR?

Thanks for your suggestion and sorry for the confusion. The numbers have been added. WT mice were exposed to IR. Main text and figure legend have been changed to make it more clear.

19. Figure 3 and legend - is the AA treatment in the figure only for TET2-OE cells? What about parental? What about n= again in this fig? MOLM cells are shown in the figure but not mentioned in the legend. For Fig. 3L it is stated in the legend that n=3 for WT but the plot seems to only show n=1?

Thank you for your questions. Control cells were treated with AA, while TET2 overexpression cell lines (TET2-OE) were not treated. We have modified the figure label, figure legend and manuscript text to clarify the fact. A-J. Data is plotted as mean with SEM (n = 3). "MOLM13" was added to the legend. Figure plot has been changed from Mean \pm SEM (TET2WT, n=3) to dot plot.

20. line 553 - sentence is cut off: "and levels of TET2..."

Thank you for pointing out this mistake. It has been fixed.

21. Supplemental figure legend S1- More detail please - # cells, methods for detecting in B and C. Label "F" is missing ahead of sentence about 14 day colony assay? No stats for part G?

Thank you for your suggestions. Detailed information has been added: A-E. Isolated mononuclear cells from *Tet2*^{+/+}, *Tet2*^{+/-}, and *Tet2*^{-/-} mice BM (B-C) and spleen (D-E) were treated at concentration of 1 million per ml with 250μM AA or H₂O. Genomic DNA was extracted after 24 hours of treatment. DNA oxidation products were assessed by dot blot (B-C) or 2D-UPLC-MS/MS (D-E). The missing "F" has been added. There is no statistically significant difference between the means of *Tet2*^{+/-} and *Tet2*^{+/-}+AA. This sentence has been added to the legend.

22. Figure S4 legend - "presence or absence of 00 uM AA" (00??)

Thank you for pointing out this mistake. It has been fixed.

23. Nuclear fractions for western blots - this reviewer does not recall where these were used. Please remind or clarify where used.

Thank you for your question. It was used in Figs. 5A and S5E.

Reviewer #2 (Remarks to the Author):

Guan and colleagues investigated the function of AA on TET2 mutant myeloid neoplasia. The authors used animal models, in vitro culture systems and biochemical analysis to demonstrate the impact of AA on Tet2. The authors employed mass spectrometry and pharmacological tools to demonstrate the effect of TET2 acetylation on AA activity. While the experiments and data are convincing, it is unclear how the author could directly connect the various acetylation level of TET2 to AA induced growth-suppressive effect in the mouse models. The molecular mechanism is not very well justified. Additional experiments are required.

Thanks for the critique of the manuscript. As per the suggestion we have performed several additional experiments and modified the text to better explain the role of AA in the TET2 activation.

1. Figure 2D-I, Tet2^{-/-}-GULO^{-/-} without AA treatment need to be shown as control.

Thank you for your question and sorry for the confusion. *GULO*^{-/-} mice (PMID: 10639167) are unable to synthesize AA, which is essential for the survival of mice. Regular chow, containing about 110 mg/kg of AA, is unable to support the growth of the mutant mice, which require AA supplemented in their drinking water. Upon withdrawal of supplementation, mice will die in weeks because of AA deficiency. In the experiment, we supplied the mice with minimal sustenance dose of AA in drinking water (0.033g/L) to maintain the mice as control and use 3.3g/L as well as 0.33g/L as concentrations for the experimental mice.

We also modified the text (lines 114-116) to make it more concise and clear.

2. Figure 3G, the labels for y-axis and x-axis are missing.

Thank you for pointing out the omission. The missing information has been added by providing the axis title.

3. The author observed increased protein level as well as the catalytic activity in TSA treated condition. It is unclear whether the increased catalytic activity is due to increased protein level or the acetylation modification could directly enhance the Tet2 catalytic activity.

Thanks for your question.

TET2 full length protein (2002 amino acid long) broadly consists of two major segments, an N-terminal regulatory domain (1-1128) and a C-terminal (1129-2002) where catalytic domain resides. Based on our work reported in this manuscript and earlier two reports (PMID: 28107650 and PMID: 30146412), acetylations of certain lysine residues are observed on both domains with opposite consequences on the TET2 dioxygenase activity. The earlier report (PMID: 28107650) demonstrated that TET2 N-terminal lysine residues K110/K111 acetylation prevents ubiquitination and proteasomal degradation leading to increased protein accumulation and enhanced activity. Consistent with this report we observed that treatment of leukemia cells by class I and II HDACs inhibitor TSA leads to higher accumulation of TET2 (Fig. 5A). This higher accumulation is also reflected in higher TET activity (Fig. 5B).

On the contrary, C-terminal lysine acetylation (K1472/1473/1478) has been reported to decrease TET2 activity (PMID: 30146412 and the present study). Consistent with this report we found that catalytic site lysine residue acetylation significantly decreases the TET2 activity that cannot be reversed by AA treatment. Treatment with either p300/CBP inhibitor that acetylate C-terminal lysine or sirtuin activator that removes lysine acetylation, restores TET2 activity that can be further amplified by AA (Figs. 5C, 5B-D and 5H). Treatment with either HAT inhibitors or sirtuin activator does not change the levels of protein (Fig. 5E-F) suggesting catalytic domain lysine residue acetylation has direct impact on the activity of the protein. These findings have been summarized in Fig. 6H.

4. Since TSA could enhance the TET2 protein level, it is unclear whether the inhibitor used in the following study (HATi, C646, SRT1720) could also affect the TET protein level. More appropriate controls are needed to clarify this point.

Thanks for your question.

In order to address this question, we performed a new experiment where, CMK cells were treated with HATi, C646, SRT1720. TET2 western blot was performed. Results demonstrate that HATi, C646 or SRT1720 does not affect levels of TET2 protein. These new data were added in the revised version (**Figs. S5E-F**).

5. Although the authors observed enhanced catalytic activity of TET2 and growth suppressive effect using pharmacology inhibitors to enhance the acetylation modification of TET2 in leukemia cells, it is unclear whether these effects are directly due to the increased acetylation on the residues mentioned in the manuscript. A further MS analysis and mutational analysis could strengthen this conclusion.

Thanks for your suggestions. We performed both analysis and presented the new set of data in **Figs. 6B-C, S4 and S6**.

Based on our analysis we identified five lysine residue mutations K1299E, K1310Q, K1491N, K1533R and K1905E in MN patients. To assess the effect of these mutations on TET2 activity we used site directed mutagenesis and generated TET2 mutants and tested its activity in HEK293T cells. The data from our observation are presented in **Figs. 6B-C and S6**.

Our data suggest that K1299E, K1310Q, K1533R and K1905E mutants express at similar levels while K1491N produced far less protein suggesting K1491N mutation may affect the TET2 protein stability (**Fig. 6B**). TET2 activity assay performed by accessing the levels of 5hmC showed that K1299E, K1310Q and K1905E mutation leads to loss of TET2 function that cannot be rescued by ascorbic acid treatment (**Figs. 6C and S6**), however, as expected K1533R mutation has no effect on TET2.

Due to the limitation of sample size the mass spectral analysis could not be performed on the primary patient bone marrow samples. However, consistent with the earlier report we expressed the full length TET2 protein in HEK293T cells and performed parallel reaction monitoring (PRM) LC-MS/MS experiments and found that K53 and K1478 are the most abundant acetylated lysine residues. We could not detect K1299, K1310, K1491, K1533 and K1905 acetylation in the present model system of HEK293T. However, based on the available structure information K1299, K1310 and K1905 are accessible for acetylation which we confirmed in a cell free system and presented as **Fig. 6D and 6G**. In addition, these residues are mutated in the several MN patients that may in part mimic consequences of catalytic domain lysine acetylation. We have modified the text to reflect our findings.

Lines 233-247:

Effect of AA on TET2 lysine mutations in myeloid neoplasia. We hypothesized that TET2 missense mutation of catalytic domain lysine residues mimicking acetylation may be found in patients with myeloid neoplasia, since acetylation leads to loss of TET2 function. TET2 has a total of 136/2002 (6.8%) lysine residues of which 55 are in the catalytic domain (residues 1129-2002). Analysis of the frequency of TET2 lysine residue mutations among 1205 *TET2^{MT}* patients (24% of 4930 patients)¹ revealed that the majority of them are frameshifts (**Table S3**). Interestingly, all 7 missense mutations (one each of K1299E, K1310Q, K1491N and K1533R and 3 K1905E) were found in the C-terminal catalytic domain (**Fig. 6A, Table S3**). In order to probe the functional consequences of these mutations, we used site directed mutagenesis and generated these missense mutations and ectopically expressed in HEK293 cells. Our data showed that TET2^{WT} and K1299E, K1310Q, K1533R and K1905E mutant proteins expressed to same levels, with exception of K1491N mutant being very lowly expressed, implicating K1491N mutation may affect the protein stability (**Fig. 6B**). The activity analysis using dot blot for 5hmC and 5mC showed that K1299E, K1310Q and K1905E mutants are completely inactive and, interestingly, addition of

ascorbic acid treatment could not restore the loss of TET2 function (**Figs. 6C and S6**).

REVIEWERS' COMMENTS:

Reviewer #1 (Remarks to the Author):

Thank you for considering and fully addressing my comments. I hope you find this has contributed to an improved manuscript. Congratulations on your very interesting study. I am hopeful this will indeed guide future therapeutic directions.

As a very minor comment, "2-oxoglutarate" is used appropriately in the manuscript but the revised abstract and introduction include a typo: "2-oxYglutarate".

Reviewer #3 (Remarks to the Author):

The authors have addressed all my previous comments.

REVIEWERS' COMMENTS:

Reviewer #1 (Remarks to the Author):

Thank you for considering and fully addressing my comments. I hope you find this has contributed to an improved manuscript. Congratulations on your very interesting study. I am hopeful this will indeed guide future therapeutic directions.

As a very minor comment, "2-oxoglutarate" is used appropriately in the manuscript but the revised abstract and introduction include a typo: "2-oxYglutarate".

Response:

We highly appreciate the critical reviews of from the reviewer. The question and comments were excellent and they help to make this manuscript better. Thank you.

We are sorry for the two typo in the manuscript. We have corrected those mistakes in the revised manuscript. Thank you.

Reviewer #3 (Remarks to the Author):

The authors have addressed all my previous comments.

We are very thankful to all your comments and suggestion during the review process.